# KNOWLEDGE LOCALIZATION: MISSION NOT ACCOMPLISHED? ENTER QUERY LOCALIZATION!

**Yuheng Chen[1,2], Pengfei Cao[1,2], Yubo Chen[1,2], Kang Liu[1,2]\*, Jun Zhao[1,2]\***

[1]The Key Laboratory of Cognition and Decision Intelligence for Complex Systems,
Institute of Automation, Chinese Academy of Sciences, Beijing, China
[2]School of Artificial Intelligence, University of Chinese Academy of Sciences, Beijing, China
chenyuheng2022@ia.ac.cn, {pengfei.cao,yubo.chen,kliu,jzhao}@nlpr.ia.ac.cn

## ABSTRACT

Large language models (LLMs) store extensive factual knowledge, but the mechanisms behind how they store and express this knowledge remain unclear. The Knowledge Neuron (KN) thesis is a prominent theory for explaining these mechanisms. This theory is based on the **Knowledge Localization** (KL) assumption, which suggests that a fact can be localized to a few knowledge storage units, namely knowledge neurons. However, this assumption has two limitations: first, it may be too rigid regarding knowledge storage, and second, it neglects the role of the attention module in knowledge expression.

In this paper, we first re-examine the KL assumption and demonstrate that its limitations do indeed exist. To address these, we then present two new findings, each targeting one of the limitations: one focusing on knowledge storage and the other on knowledge expression. We summarize these findings as **Query Localization** (QL) assumption and argue that the KL assumption can be viewed as a simplification of the QL assumption. Based on QL assumption, we further propose the Consistency-Aware KN modification method, which improves the performance of knowledge modification, further validating our new assumption. We conduct 39 sets of experiments, along with additional visualization experiments, to rigorously confirm our conclusions. Code is available here.

## 1 INTRODUCTION

Large language models (LLMs) are believed to store extensive factual knowledge (MetaAI, 2024; Touvron et al., 2023), however, the mechanisms behind this storage and expression have not been well-explained. The Knowledge Neurons (KN) thesis (Dai et al., 2022; Meng et al., 2022; 2023; Niu et al., 2024; Chen et al., 2024b;a) is a prominent theory aiming to explain these mechanisms. It proposes that LLMs recall facts through their multi-layer perceptron (MLP) weights, referring to the units responsible for storing knowledge as knowledge neurons (KNs). Based on this, KN-inspired model editing methods are proposed (Meng et al., 2022; 2023), which first localize knowledge neurons and then modify them to update knowledge, providing further support for the KN thesis. Not only them, but also many works have adopted KN theory and applied it to study downstream tasks (Chen et al., 2024b;a; Wang et al., 2024c), making its theoretical foundation crucial.

In fact, the KN thesis is based on the knowledge localization (**KL**) assumption: a piece of factual knowledge can be localized to several knowledge neurons. However, this assumption has two limitations. (1) In terms of knowledge storage, if we refer to different rephrased queries expressing the same fact as *neighbor queries*, and the corresponding knowledge neurons as *neighbor KNs*, then the KL assumption implies that neighbor KNs are consistent. However, as Figure 1 illustrates, while the neighbor KNs of $Fact_1$ exhibit high consistency, those of $Fact_2$ show low consistency, indicating the KL assumption does not hold universally. We denote facts that satisfy the KL assumption as **Consistent Knowledge** ($K_C$, e.g., $Fact_1$), while facts that violate the KL assumption are categorized as **Inconsistent Knowledge** ($K_I$, e.g., $Fact_2$). Previous research and the KL assumption essentially assume that all factual knowledge belongs to $K_C$. (2) In terms of knowledge expression, the KL

---

\*Corresponding authors.

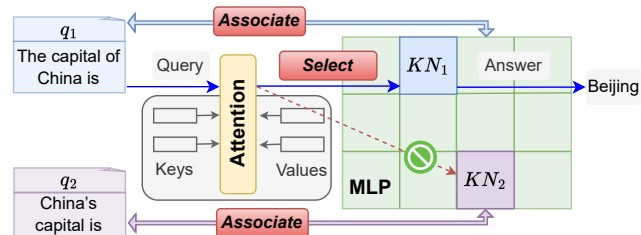

Figure 1: Heatmaps of the neuron activation values, with darker colors indicating higher values (can be viewed as knowledge neurons). The left two heatmaps show neuron activations for two neighbor queries of ⟨*Suleiman I, position, Shah*⟩ (Fact$_1$), while the right two correspond to ⟨*Christoph Ahlhaus, position, mayor*⟩ (Fact$_2$).

assumption overlooks the attention module, yet there must be interconnections between the different modules in LLMs. Similarly, since KL only considers the role of the MLP module in storing knowledge, it does not take into account how the model selects and expresses this knowledge to answer queries. Therefore, we re-examine the KL assumption and raise questions Q1 and Q2:

**Q1**: Does the KL assumption hold for all facts? If not, is $K_I$ widely prevalent? (§2)

**A1** We investigate the knowledge localization assumption and find that the universal presence of $K_I$ that violates this assumption.

(1) **Statistical Evidence.** As shown in Figure 1, if the knowledge neurons corresponding to a fact exhibit low consistency for its neighbor queries, it indicates that the fact does not conform to the KL assumption. Based on this observation, we propose a metric to evaluate the consistency among neighbor KNs, and the statistical results show that a significant proportion of facts belong to $K_I$. For example, in LLaMA3-8b, this proportion reaches 77%. This directly proves that facts that do not conform to the KL assumption are widespread.

Figure 2: The Query Localization assumption.

(2) **Modification-Based Evidence.** We categorize facts into $K_C$ and $K_I$ based on their consistency scores to perform knowledge erasure and updates. We find that for facts in $K_I$, editing the KNs corresponding to the query itself does not generalize well to neighbor queries. This indirectly indicates that the neighbor KNs for $K_I$ are inconsistent. In summary, the answer to Q1 is: the KL assumption is not always valid and $K_I$ is widely prevalent.

**Q2**: Since the KL assumption has two limitations, what is a more realistic assumption? (§3)

**A2** Our two findings address the two limitations of the knowledge localization assumption.

(1) **Query-KN Mapping**: In terms of knowledge storage, the KL assumption implies that localization results are static and universally applicable across all queries. However, our findings indicate that for facts in $K_I$, localization results are influenced by the query context rather than being fixed. In other words, knowledge neurons are associated with the query rather than the fact. For instance, Figure 1 shows that different neighbor queries for Fact$_2$ correspond to different knowledge neurons. Similarly, in Figure 2, neighbor queries $q_1$ and $q_2$ are associated with distinct KNs ($KN_1$ and $KN_2$).

(2) **Dynamic KN Selection**. In terms of knowledge expression, the KL assumption overlooks the role of the attention module. Our findings show that LLMs rely on the attention module to select appropriate KNs to answer a specific query. For example, in Figure 2, neighbor queries $q_1$ and $q_2$ are associated with different KNs. Then, when $q_1$ is input, $KN_1$ is activated and selected to provide the answer "Beijing", while the activation value of $KN_2$ remains low, preventing it from being selected.

Based on these insights, we propose the **Query Localization (QL)** assumption, which consists of query-KN mapping and dynamic KN selection. To further demonstrate the validity of our assumption, we apply it in model editing experiments. We propose the Consistency-Aware KN modification

method, which leverages the QL assumption to improve knowledge modification, achieving an 8% and 9% performance improvement over two baselines in the "Erasure" setting on LLaMA3-8b, further validating the QL assumption. In summary, the answer to Q2 is: a more realistic assumption is the Query Localization assumption. Our contributions are summarized as follows:

- We conduct the first in-depth exploration of the Knowledge Localization assumption, a foundational and widely accepted assumption. We classify facts into $K_C$ and $K_I$, and demonstrate that $K_I$, i.e., facts that do not adhere to this assumption, are widely present.
- We propose a more realistic Query Localization assumption, which includes two parts: query-KN mapping and dynamic KN selection. This addresses the limitations of the KL assumption in both knowledge storage and expression.
- We apply the QL assumption to improve knowledge modification methods, further validating the soundness of the QL assumption.

## 2 EXPLORING KNOWLEDGE LOCALIZATION LIMITATIONS

This section investigates Q1 and demonstrates the existence of Inconsistent Knowledge ($K_I$), which does not satisfy the knowledge localization (KL) assumption. Our experiments adopt GPT-2 (Radford et al., 2019), LLaMA2-7b (Touvron et al., 2023), and LLaMA3-8b (MetaAI, 2024), representing a range of sizes of popular auto-regressive models. This allows us to assess the scalability of our methods and conclusions. Consistent with other knowledge localization methods (Dai et al., 2022; Chen et al., 2024a), we employ the fill-in-the-blank cloze task (Petroni et al., 2019) to assess whether a pretrained model knows a fact. Regarding the dataset, we employ the ParaRel dataset (Elazar et al., 2021). For details to the dataset, see Table 5 in Appendix B.

### 2.1 STATISTICAL EVIDENCE FOR THE EXISTENCE OF INCONSISTENT KNOWLEDGE

In this subsection, we prove that the consistency of knowledge neurons of some facts is very low, which shows that these facts do not conform to the knowledge localization assumption.

**Consistency Analysis** According to the KL assumption, neighbor queries should be localized to the same KNs, with any deviations primarily attributable to the localization method itself. To assess this, we calculate the corresponding KNs for each query and introduce the KN-Consistency Score (CS) metric. Given a fact with $k$ neighbor queries $\{q_1, \ldots, q_k\}$, we calculate its CS as follows:

$$CS_{\text{orig}} = \frac{\left| \bigcap_{i=1}^{k} \mathcal{N}_i \right|}{\left| \bigcup_{i=1}^{k} \mathcal{N}_i \right|} \xrightarrow{\text{relaxation}} CS = \frac{\left| \left\{ n \mid \sum_{i=1}^{k} \mathbf{1}_{n \in \mathcal{N}_i} > 1 \right\} \right|}{\left| \bigcup_{i=1}^{k} \mathcal{N}_i \right|} \quad (1)$$

where $\mathcal{N}_i$ is the set of knowledge neurons corresponding to query $q_i$, and $n$ denote the knowledge neuron. $\mathbf{1}_{n \in \mathcal{N}_i}$ is an indicator function, which equals 1 if $n$ belongs to $\mathbf{1}_{n \in \mathcal{N}_i}$. Thus, $\sum_{i=1}^{k} \mathbf{1}_{n \in \mathcal{N}_i}$ represents the number of times $n$ appears across all KN sets (i.e., $\mathcal{N}_i$). In the original metric, $CS_{\text{orig}}$, the numerator represents the intersection of all $\mathcal{N}_i$, meaning a KN must appear in all sets to be counted. After relaxation ($CS$), the numerator includes any KN that appears in more than one of the $\mathcal{N}_i$ sets, allowing it to be counted even if it is not present in every set. This relaxation reduces the impact of localization errors and provides stronger evidence for the existence of $K_I$.

Then, we use a thresholding technique based on $CS$, classifying facts above a certain threshold as $K_C$ (consistent knowledge) and those below it as $K_I$ (inconsistent knowledge). We consider two types of thresholds: a static threshold and Otsu's threshold[1]. While Otsu's threshold aims to maximize the between-class variance and effectively separate two classes of data, the static threshold reflects the inherent nature of a fact's adherence (or non-adherence) to the KL assumption. See Table 4 in A for specific thresholds. To ensure our findings are not method-specific, we compare three advanced knowledge localization methods (Dai et al., 2022; Enguehard, 2023; Chen et al., 2024a), with minor modifications for task adaptation, primarily to the method of Enguehard (2023) (detailed in Appendix D). Finally, we apply Welch's t-test[2] to confirm the statistical significance of the difference between $K_C$ and $K_I$.

---

[1] https://en.wikipedia.org/wiki/Otsu%27s_method
[2] https://en.wikipedia.org/wiki/Welch%27s_t-test

| T | GPT-2 Dai et al. (2022) | | | | | Enguehard (2023) | | | | | Chen et al. (2024a) | | | | | $U_I$ |
|---|---|---|---|---|---|---|---|---|---|---|---|---|---|---|---|---|
| | $R_C$ | $CS_C$ | $R_I$ | $CS_I$ | $t$ | $R_C$ | $CS_C$ | $R_I$ | $CS_I$ | $t$ | $R_C$ | $CS_C$ | $R_I$ | $CS_I$ | $t$ | |
| St | 0.56 | 0.21 | **0.44** | 0.03 | 236 | 0.54 | 0.23 | **0.46** | 0.03 | 235 | 0.53 | 0.25 | **0.47** | 0.03 | 230 | **0.42** |
| Ot | 0.41 | 0.24 | **0.59** | 0.06 | 223 | 0.44 | 0.29 | **0.55** | 0.05 | 219 | 0.40 | 0.29 | **0.60** | 0.06 | 221 | **0.53** |

| T | LLaMA2-7b Dai et al. (2022) | | | | | Enguehard (2023) | | | | | Chen et al. (2024a) | | | | | $U_I$ |
|---|---|---|---|---|---|---|---|---|---|---|---|---|---|---|---|---|
| | $R_C$ | $CS_C$ | $R_I$ | $CS_I$ | $t$ | $R_C$ | $CS_C$ | $R_I$ | $CS_I$ | $t$ | $R_C$ | $CS_C$ | $R_I$ | $CS_I$ | $t$ | |
| St | 0.40 | 0.21 | **0.60** | 0.04 | 158 | 0.39 | 0.20 | **0.61** | 0.04 | 150 | 0.40 | 0.20 | **0.60** | 0.04 | 160 | **0.55** |
| Ot | 0.21 | 0.28 | **0.79** | 0.062 | 152 | 0.20 | 0.25 | **0.80** | 0.07 | 158 | 0.24 | 0.30 | **0.76** | 0.06 | 132 | **0.70** |

| T | LLaMA3-8b Dai et al. (2022) | | | | | Enguehard (2023) | | | | | Chen et al. (2024a) | | | | | $U_I$ |
|---|---|---|---|---|---|---|---|---|---|---|---|---|---|---|---|---|
| | $R_C$ | $CS_C$ | $R_I$ | $CS_I$ | $t$ | $R_C$ | $CS_C$ | $R_I$ | $CS_I$ | $t$ | $R_C$ | $CS_C$ | $R_I$ | $CS_I$ | $t$ | |
| St | 0.16 | 0.16 | **0.84** | 0.03 | 114 | 0.15 | 0.18 | **0.85** | 0.03 | 105 | 0.18 | 0.19 | **0.82** | 0.03 | 123 | **0.77** |
| Ot | 0.23 | 0.14 | **0.77** | 0.03 | 128 | 0.21 | 0.15 | **0.79** | 0.03 | 107 | 0.24 | 0.16 | **0.76** | 0.03 | 130 | **0.70** |

Table 1: Overall results of Consistency Analysis. The symbol **T** represents the static (St) and Otsu (Ot) thresholds. The $t$-statistics and $p$-values are from the T-test, with $p < 1e - 6$ in all cases.

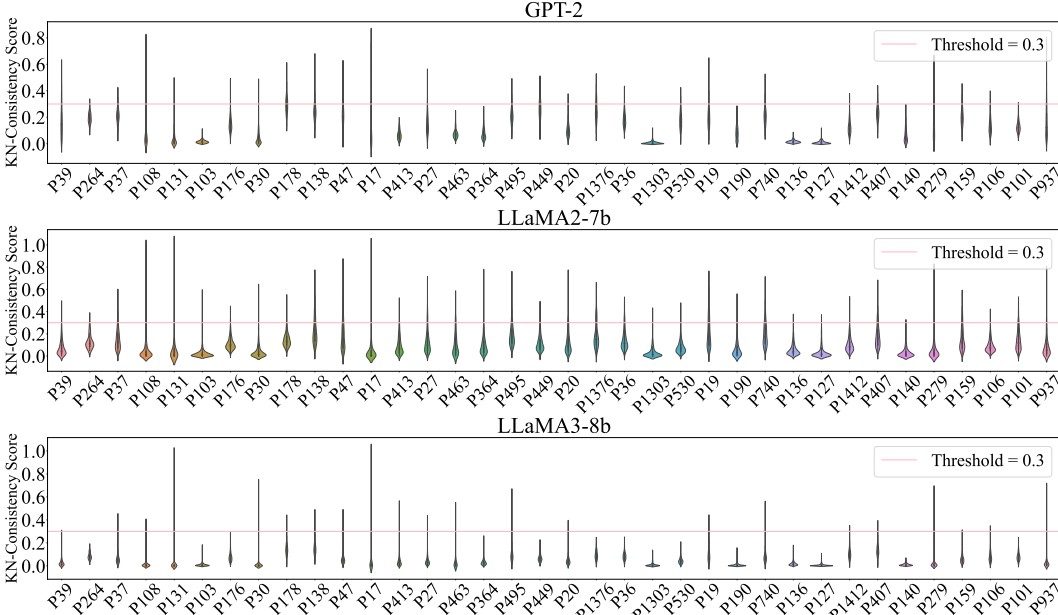

Figure 3: Violin plot for Consistency Analysis. The $x$-axis are the fact relations, and the $y$-axis is the $CS_2$ value. The width of each violin plot indicates the density of data at different $CS_2$ values. We select a threshold of 0.3 as an example, and facts below this threshold are classified as $K_I$.

Regarding the evaluation metrics, we calculate the proportions of $K_C$ and $K_I$, denoted as $R_C$ and $R_I$, respectively. We also compute the average values of $CS$ for these facts, denoted as $CS_C$ and $CS_I$. Furthermore, we calculate the proportion of facts classified as $K_I$ by all three methods, denoted as $U_I$ (i.e., the union of $K_I$).

**Findings** Figure 3 classifies facts based on their respective relations (e.g., P39 represents the "position" relation), illustrating the distribution of $CS$ when utilizing the knowledge localization method proposed by Dai et al. (2022). The violin plots for other methods can be found in Figures 7 and 8 in Appendix C. Together, Figure 3 and Table 1 summarize the overall results.

(1) Inconsistent knowledge ($K_I$) is widely present across different knowledge localization methods, LLMs, and relations. In Table 1, the consistently high ratio of $K_I$ ($R_I$) and low $CS$ values ($CS_I$) demonstrate that the proportion of facts categorized as $K_I$ is substantial across different methods,

with $U_I$ showcasing high classification agreement among all three knowledge localization methods. For LLaMA3, using a static threshold, 77% of the facts are consistently classified into $K_I$. Moreover, in Figure 3, using an example threshold of 0.3, the majority of facts across various relations fall below this threshold, thus belonging to $K_I$. (2) Statistical tests reveal a significant difference between $K_C$ and $K_I$. For instance, using the static threshold (St) for LLaMA3-8b, the recorded $t$-statistic is 123, with a $p$-value less than $1e - 6$. These results reflect a very strong distinction, as the high $t$-statistic and extremely low $p$-value show that the difference is highly reliable. Combining (1) and (2), we conclude that **inconsistent knowledge ($K_I$) is prevalent**. Beyond statistical analysis, we further validate the existence of KI through knowledge modification experiments.

## 2.2 Modification-Based Evidence for the Existence of Inconsistent Knowledge

In this subsection, we conduct knowledge modification experiments to demonstrate the existence of inconsistent knowledge ($K_I$). We use a static threshold to classify facts into $K_C$ and $K_I$.

**Experimental setups**  Let $\langle s, r, o \rangle$ denote a fact consisting of a subject ($s$), relation ($r$), and object ($o$). We perform two types of knowledge modification: Erasure and Update. Given a fact with $k$ queries $\{q_1, \ldots, q_k\}$, and for a query $q_i$, modify the MLP weights of LLMs as follows.

$$W_{l,p} = \begin{cases} 0, & \text{if Erasure} \\ W_{l,p} - \lambda_1 E(o) + \lambda_2 E(o'), & \text{if Update} \end{cases} \tag{2}$$

where $l$ and $p$ represent the layer and position of the knowledge neuron, $W_{l,p}$ is the corresponding MLP weight. $E(o)$ and $E(o')$ are the word embeddings of the original object $o$ and the updated object $o'$, respectively. $\lambda_1$ and $\lambda_2$ are hyperparameters. We perform knowledge modification on two different KN sets: (1) $\mathcal{N}_i$, the set of knowledge neurons corresponding to $q_i$, (2) $\mathcal{N}_u$, the union of KNs across all $k$ queries, i.e., $\mathcal{N}_u = \bigcup_{i=1}^{k} \mathcal{N}_i$.

**Evaluation Metrics**  (1) *Knowledge Modification Metrics*: We adopt three metrics (detailed in E): Reliability (Rel), Generalization (Gen), and Locality (Loc) (Yao et al., 2023). These three metrics respectively represent the model's ability to answer the original query, neighbor queries, and unrelated queries after knowledge modification. All three metrics are better when higher. To facilitate comparison, we also calculate the average of these three metrics (Avg).

(2) *General Capability Metrics*: Editing neurons may disrupt the model's performance in generating text (Zhang et al., 2024; Zhao et al., 2023). Similar to other model editing methods (Wang et al., 2024b), we employ the perplexity (PPL) metric to evaluate the model's general capability after modification. Specifically, we randomly select five entries from WikiText2 (Merity et al., 2017) each time and calculate the relative increase in PPL before (b) and after (a) editing the model: $\Delta\text{PPL} = \frac{\text{PPL}_a - \text{PPL}_b}{\text{PPL}_b}$. A lower $\Delta\text{PPL}$ is better, as it indicates less disruption to the model.

**Findings**  Table 2 presents the results of this experiment, leading us to the following conclusions.

(1) **Low Generalization in Inconsistent Knowledge in $\mathcal{N}_i$**: Modifying $\mathcal{N}_i$, i.e., the KNs corresponding to $q_i$, leads to low generalization for $K_I$. Specifically, under the "Erasure" setting, the generalization scores are only 0.09 for GPT-2 and 0.04 for LLaMA3-8b, indicating unsuccessful modification of neighbor queries. Despite high Reliability and Locality scores on original and unrelated queries, the poor generalization reveals the limitations of this method. In contrast, $K_C$ exhibits higher "Avg" and "Gen" metrics. For example, for LLaMA3, "Avg" and "Gen" metric values reach 0.47 and 0.30, respectively, suggesting better consistency among neighbor KNs (i.e., KNs corresponding to neighbor queries).

(2) **High $\Delta\text{PPL}$ and lower Locality for Inconsistent Knowledge in $\mathcal{N}_u$**: To achieve high generalization for $K_I$, substantial modifications to $\mathcal{N}_u$ (union of $\mathcal{N}_i$) are required, necessitating the alteration of many KNs to impact a single fact. However, this approach significantly increases perplexity change ($\Delta\text{PPL}$), with a peak of 1.05 for LLaMA3-8b under the "Erasure" setting (i.e., a 105% increase in PPL), and causes Locality to drop from 0.80 to 0.50, indicating excessive alterations to model parameters. It is precisely because the neighbor KNs are inconsistent that taking

| $\mathcal{N}_i$ | *Erasure* | | | | | | | | | | | | | | | |
|---|---|---|---|---|---|---|---|---|---|---|---|---|---|---|---|
| | **GPT-2** | | | | | **LLaMA2-7b** | | | | | **LLaMA3-8b** | | | | |
| | Rel | Gen | Loc | **Avg** | $\Delta$ **PPL** | Rel | Gen | Loc | **Avg** | $\Delta$ **PPL** | Rel | Gen | Loc | **Avg** | $\Delta$ **PPL** |
| $K_C$ | 0.55 | 0.47 | 0.93 | 0.65 | 0.02 | 0.33 | 0.34 | 0.79 | 0.49 | 0.01 | 0.28 | 0.30 | 0.83 | 0.47 | 0.04 |
| $K_I$ | 0.50 | **0.09** | 0.97 | 0.52 | 0.06 | 0.36 | **0.11** | 0.80 | 0.42 | 0.03 | 0.34 | **0.04** | 0.90 | 0.43 | 0.05 |

| $\mathcal{N}_u$ | *Erasure* | | | | | | | | | | | | | | | |
|---|---|---|---|---|---|---|---|---|---|---|---|---|---|---|---|
| | **GPT-2** | | | | | **LLaMA2-7b** | | | | | **LLaMA3-8b** | | | | |
| | Rel | Gen | Loc | **Avg** | $\Delta$ **PPL** | Rel | Gen | Loc | **Avg** | $\Delta$ **PPL** | Rel | Gen | Loc | **Avg** | $\Delta$ **PPL** |
| $K_C$ | 0.58 | 0.55 | 0.90 | 0.68 | 0.12 | 0.30 | 0.55 | 0.70 | 0.52 | 0.08 | 0.30 | 0.35 | 0.80 | 0.48 | 0.18 |
| $K_I$ | 0.65 | 0.60 | 0.70 | 0.65 | **2.02** | 0.44 | 0.45 | 0.52 | 0.42 | **1.50** | 0.36 | 0.40 | 0.50 | 0.49 | **1.05** |

| $\mathcal{N}_i$ | *Update* | | | | | | | | | | | | | | | |
|---|---|---|---|---|---|---|---|---|---|---|---|---|---|---|---|
| | **GPT-2** | | | | | **LLaMA2-7b** | | | | | **LLaMA3-8b** | | | | |
| | Rel | Gen | Loc | **Avg** | $\Delta$ **PPL** | Rel | Gen | Loc | **Avg** | $\Delta$ **PPL** | Rel | Gen | Loc | **Avg** | $\Delta$ **PPL** |
| $K_C$ | 0.53 | 0.40 | 0.99 | 0.64 | 0.04 | 0.30 | 0.39 | 0.89 | 0.53 | 0.03 | 0.30 | 0.29 | 0.79 | 0.46 | 0.07 |
| $K_I$ | 0.44 | **0.11** | 0.96 | 0.50 | 0.09 | 0.39 | **0.07** | 0.80 | 0.42 | 0.08 | 0.29 | **0.08** | 0.86 | 0.41 | 0.08 |

| $\mathcal{N}_u$ | *Update* | | | | | | | | | | | | | | | |
|---|---|---|---|---|---|---|---|---|---|---|---|---|---|---|---|
| | **GPT-2** | | | | | **LLaMA2-7b** | | | | | **LLaMA3-8b** | | | | |
| | Rel | Gen | Loc | **Avg** | $\Delta$ **PPL** | Rel | Gen | Loc | **Avg** | $\Delta$ **PPL** | Rel | Gen | Loc | **Avg** | $\Delta$ **PPL** |
| $K_C$ | 0.56 | 0.48 | 0.88 | 0.64 | 0.13 | 0.32 | 0.59 | 0.82 | 0.68 | 0.10 | 0.44 | 0.41 | 0.74 | 0.53 | 0.22 |
| $K_I$ | 0.54 | 0.55 | 0.74 | 0.61 | **1.88** | 0.40 | 0.43 | 0.62 | 0.48 | **1.16** | 0.29 | 0.33 | 0.66 | 0.43 | **0.93** |

Table 2: Results of Knowledge Modification. $\mathcal{N}_i$ and $\mathcal{N}_u$ are the two sets of knowledge neurons. The bolded results indicate poor performance, reflecting **Failures** in model editing.

the intersection (i.e., $\mathcal{N}_u$) results in a large number of neurons. This observation suggests that facts related to $K_I$ cannot be localized to a fixed set of KNs. Together, findings (1) and (2) confirm that $K_I$ does not adhere to the KL assumption.

**Let's review Q1, we have demonstrated, from both statistical and model editing perspectives, that the knowledge localization assumption does not always hold.** Next, we explore a more realistic alternative.

## 3 QUERY LOCALIZATION ASSUMPTION

**Motivation** Since inconsistent knowledge ($K_I$) does not satisfy the KL assumption, we naturally raise the question Q2: What is a more realistic assumption? Let's revisit the two limitations with the knowledge localization assumption: (1) Knowledge neurons correspond to facts, meaning that a fact is statically stored by several KNs. However, Tables 1 and 2 suggest that this assumption does not hold for $K_I$. (2) The focus has been solely on the MLP module, neglecting the attention module. In light of recent research on the attention module (Ren et al., 2024; Geva et al., 2023; Meng et al., 2022), we argue that it should also be considered. Below, we will explain our two findings that address these limitations, including Query-KN Mapping (§3.1) and Dynamic KN Selection (§3.2).

### 3.1 QUERY-KN MAPPING

**Method** In order to explore the relationship between queries and knowledge neurons, we manipulate the KN activation values by either suppressing or enhancing them. As before, we first classify facts into inconsistent knowledge ($K_I$) and consistent knowledge ($K_C$). Then, given a fact with $k$ queries $\{q_1, \ldots, q_k\}$, and for a specific query $q_i$, we manipulate five different sets of neurons and study how such operations affect the model's response to the query:

(1) Self: Equivalent to $\mathcal{N}_i$, the set of KNs corresponding to $q_i$. (2) Union: The union of other neighbor KNs, i.e., the union of KNs corresponding to the neighbor queries. (3) Intersection: The intersection of other neighbor KNs. (4) Refine: Refined neighbor KNs, the set of KNs that appear more than once. (5) Unrelated: Randomly selected unrelated neurons, equal in number to $\mathcal{N}_i$.

Regarding evaluation metrics, we follow other knowledge localization methods (Dai et al., 2022; Chen et al., 2024a;b), and calculate the rates of increase and decrease in the LLMs' answer proba-

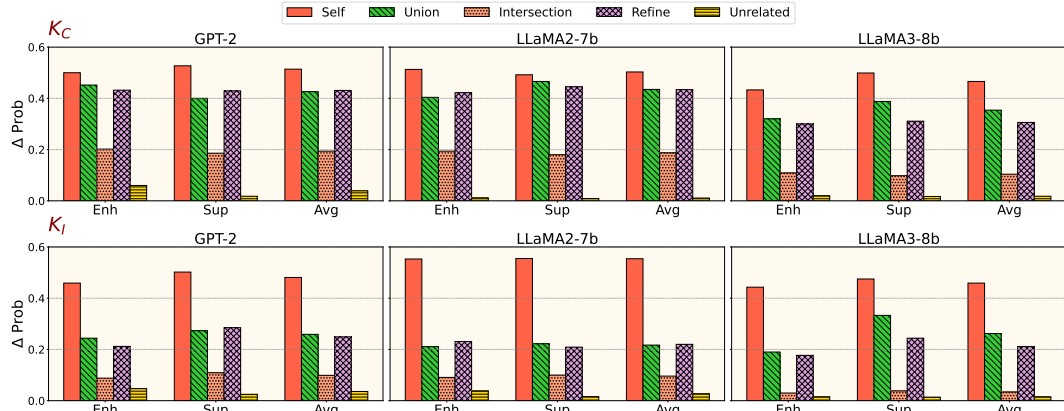

Figure 4: Results of Query-KN Mapping. "Enh" and "Sup" refer to enhancement and suppression of KN activation values, respectively, with "Avg" representing their average.

bilities before (b) and after (a) suppression and enhancement: $\Delta\text{Prob} = \pm\frac{\text{Prob}_a - \text{Prob}_b}{\text{Prob}_b}$. Here, "$\pm$" indicates that we assign a negative value for suppression and a positive value for enhancement.

**Findings** Figure 4 illustrates our results, leading to three findings. (1) Regardless of whether it is inconsistent knowledge ($K_I$) and consistent knowledge ($K_C$), the results from the $\mathcal{N}_i$ indicate that the influence of suppressing or enhancing the query's own KNs is significant. This suggests a strong association between the KNs and the query. (2) For $K_I$, the average values ("Avg") of other baselines are considerably lower than those of $\mathcal{N}_i$, particularly in the "Intersection" case. In contrast, for $K_C$, the average values decrease to a lesser extent. This indicates that the neighbor KNs for $K_C$ are more consistent, while the neighbor KNs for $K_I$ exhibit much less consistency. This demonstrates that for $K_I$, the KNs do not correspond to the fact itself.

Combining (1) and (2), we conclude the phenomenon of **Query-KN Mapping: For inconsistent knowledge, the localization results are associated with the query rather than the fact.**

(3) Furthermore, for $K_C$, the values also decrease. This is because we use a very low threshold when classifying facts to rigorously demonstrate the presence of $K_I$. As a result, some facts in $K_C$ may not be entirely consistent, which actually strengthens our conclusion by confirming that $K_I$ exists. Therefore, $K_C$ can be considered a special case of $K_I$.

## 3.2 DYNAMIC KN SELECTION

**Methods** To address the issue of the knowledge localization assumption neglecting the attention module, we employ a method similar to manipulating neuron values by suppressing or enhancing attention scores, thereby exploring their effects. Notably, attention score matrices resemble KN activation value matrices, differing only in an additional dimension for attention heads. Thus, similar to the definition of knowledge neurons, we identify column vectors with high attention scores. Drawing inspiration from cognitive science (Dalva et al., 2007; Kim et al., 2018; Lisman et al., 2018; Harikesh et al., 2022; Rabinowitch et al., 2024), we refer to these vectors as *Knowledge Synapses* (**KS**), denoted as $\mathcal{S}$. Given a query with its answer, the KSs are defined as:

$$\tau = \alpha \cdot \frac{1}{C \cdot L \cdot H} \sum_{l=1}^{L} \sum_{h=1}^{H} \sum_{r=1}^{R} \sum_{c=1}^{C} A_{(l,h)}^{(r,c)} \tag{3}$$

$$\mathcal{S} = \left\{ (c,l,h) \mid \sum_{r=1}^{R} A_{(l,h)}^{(r,c)} > \tau, \ \forall \, l \in \{1,\ldots,L\}, h \in \{1,\ldots,H\}, c \in \{1,\ldots,C\} \right\} \tag{4}$$

where $\tau$ is the dynamic threshold, $\alpha$ is a scaling factor, and $A$ is the attention score matrix. $L$ and $H$ represent the number of layers and heads of the attention module, respectively, while $R$ and $C$ are the rows and columns of $A$. $l$, $h$, $r$, $c$ are the corresponding indices. After localizing $\mathcal{S}$, we enhance or suppress the attention scores at these positions (i.e., KS attention scores) to study their effects.

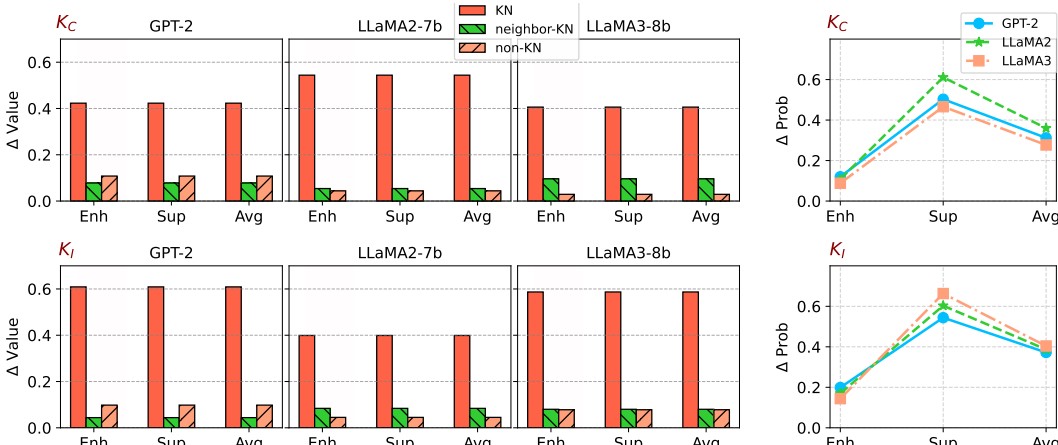

Figure 5: Heatmaps showing the neuron activation values, after suppressing knowledge synapses. The queries used here are the same as those in Figure 1. The dark areas in Figure 1 appear lighter here, indicating a decrease in the activation value of knowledge neurons. For the enhanced case, see Figure 10 in Appendix C.

Figure 6: Results of Dynamic KN Selection. "Avg" represents the average between the "Enh" and "Sup" settings.

**Evaluation Metrics and Baselines**   (1) We assess the impact of suppressing (Sup) or enhancing (Enh) knowledge synapses on the activation values of knowledge neurons. We calculate the rates of increase and decrease in the average KN activation values before (b) and after (a) KS manipulation: $\Delta\text{Value} = \pm\frac{\text{Value}_a - \text{Value}_b}{\text{Value}_b}$. (2) We assess the impact of KSs on knowledge expression by computing the change in the LLMs' answer probability ($\Delta$Prob).

To further demonstrate the "selection" role of the attention module, we compare the changes in values of neurons from two other sets: (1) neighbor KNs, i.e., the knowledge neurons corresponding to the neighbor queries, and (2) non-KNs, i.e., randomly selected non-knowledge neurons.

**Findings**   Figure 5 illustrates a case, while Figure 6 presents the overall results, revealing three key phenomena: (1) Manipulating KSs leads to a higher $\Delta$Value for knowledge neurons and also has a noticeable impact on $\Delta$Prob (i.e., the answer probability). (2) When manipulating KSs, whether they are neighbor KNs or non-KNs, the corresponding KN-values do not show significant changes. (3) Manipulating knowledge synapses significantly affects both $K_C$ and $K_I$.

Combined with Figure 5, we conclude that suppressing attention scores significantly decreases KN-values, while the value changes in other neurons are relatively minor, regardless of the knowledge category. This hinders the model's ability to select appropriate KNs for accurate knowledge expression, resulting in a decrease in LLMs' answer probabilities. We summarize this phenomenon as **Dynamic KN Selection: the attention module plays a role in selecting specific KNs for expressing knowledge.**

(4) Additionally, under the "Enh" setting, $\Delta$Prob is significantly lower than under "Sup". This is because, without suppressing KSs, the attention module is already capable of selecting KNs. Further enhancement leads to a "saturation" effect, where the accuracy of KN selection reaches its limit. This further proves that the attention module plays a selective role rather than a storage role.

| Method | GPT-2 | | | | | LLaMA2-7b | | | | | LLaMA3-8b | | | | |
|---|---|---|---|---|---|---|---|---|---|---|---|---|---|---|---|
| | | | | | | | | | | *Erasure* | | | | | |
| | Rel | Gen | Loc | **Avg** | **Δ PPL** | Rel | Gen | Loc | **Avg** | **Δ PPL** | Rel | Gen | Loc | **Avg** | **Δ PPL** |
| $K_C$ ($\mathcal{N}_i$) | 0.55 | 0.47 | 0.92 | 0.64 | **0.02** | 0.33 | 0.34 | 0.79 | 0.49 | **0.01** | 0.28 | 0.30 | 0.83 | 0.47 | 0.04 |
| $K_C$ ($\mathcal{N}_u$) | 0.58 | 0.55 | 0.90 | **0.68** | 0.12 | 0.30 | 0.55 | 0.70 | 0.52 | 0.08 | 0.30 | 0.35 | 0.81 | **0.49** | 0.18 |
| $K_C$ (Ours) | 0.56 | 0.50 | 0.90 | 0.65 | 0.03 | 0.32 | 0.44 | 0.88 | **0.55** | 0.03 | 0.30 | 0.33 | 0.80 | 0.48 | **0.02** |
| $K_I$ ($\mathcal{N}_i$) | 0.50 | 0.09 | 0.97 | 0.52 | **0.06** | 0.36 | 0.11 | 0.80 | 0.42 | **0.03** | 0.34 | 0.04 | 0.90 | 0.43 | **0.05** |
| $K_I$ ($\mathcal{N}_u$) | 0.65 | 0.60 | 0.70 | **0.65** | 2.02 | 0.44 | 0.45 | 0.52 | 0.47 | 1.50 | 0.36 | 0.40 | 0.50 | 0.42 | 1.05 |
| $K_I$ (Ours) | 0.55 | 0.40 | 0.90 | 0.62 | 0.10 | 0.35 | 0.35 | 0.77 | **0.49** | 0.06 | 0.34 | 0.30 | 0.88 | **0.51** | 0.09 |
| | | | | | | | | | | *Update* | | | | | |
| | Rel | Gen | Loc | **Avg** | **Δ PPL** | Rel | Gen | Loc | **Avg** | **Δ PPL** | Rel | Gen | Loc | **Avg** | **Δ PPL** |
| $K_C$ ($\mathcal{N}_i$) | 0.53 | 0.40 | 0.99 | 0.64 | **0.04** | 0.30 | 0.39 | 0.89 | 0.53 | **0.03** | 0.30 | 0.29 | 0.79 | 0.46 | **0.07** |
| $K_C$ ($\mathcal{N}_u$) | 0.56 | 0.48 | 0.88 | 0.64 | 0.13 | 0.32 | 0.59 | 0.82 | **0.68** | 0.10 | 0.44 | 0.41 | 0.74 | **0.53** | 0.22 |
| $K_C$ (Ours) | 0.55 | 0.44 | 0.98 | **0.66** | 0.14 | 0.30 | 0.50 | 0.88 | 0.56 | 0.10 | 0.35 | 0.33 | 0.70 | 0.46 | 0.10 |
| $K_I$ ($\mathcal{N}_i$) | 0.44 | 0.11 | 0.96 | 0.50 | **0.09** | 0.39 | 0.07 | 0.80 | 0.42 | **0.08** | 0.29 | 0.08 | 0.86 | 0.41 | **0.08** |
| $K_I$ ($\mathcal{N}_u$) | 0.54 | 0.55 | 0.74 | **0.61** | 1.88 | 0.40 | 0.43 | 0.62 | 0.48 | 1.16 | 0.29 | 0.33 | 0.66 | 0.43 | 0.93 |
| $K_I$ (Ours) | 0.45 | 0.45 | 0.88 | 0.59 | 0.20 | 0.40 | 0.38 | 0.75 | **0.51** | 0.12 | 0.29 | 0.29 | 0.80 | **0.46** | 0.14 |

Table 3: Results of Consistency-Aware KN Modification. The best results are indicated in **bold**, and the second-best results are indicated with underline. Align with Table 2, the higher the "Avg" the better, the lower the "Δ PPL" the better.

**Let's review Q2, and our two findings address the two limitations of the KL assumption. We summarize our findings to establish a more realistic *Query Localization* assumption, which includes Query-KN Mapping and Dynamic KN Selection.**

## 3.3 APPLICATION OF QL ASSUMPTION: CONSISTENCY-AWARE KN MODIFICATION

**Experimental Setups** Inspired by query-KN mapping, we propose a new approach to select knowledge neurons that improves knowledge modification methods. By incorporating KN consistency, we introduce the (CAS) metric, which penalizes low consistency and rewards high activation values. Given a fact with $k$ queries $\{q_1, \ldots, q_k\}$, for query $q_i$, the CAS for the $p$-th neuron in the $l$-th layer is defined as:

$$CAS_{(l,p)} = \beta_1 \mu_{lp} - \beta_2 \sigma_{lp}, \quad \text{where } \mu_{lp} = \frac{1}{k}\sum_{i=1}^{k} as_{lp}^{(i)}, \quad \sigma_{lp} = \sqrt{\frac{1}{k}\sum_{i=1}^{k}(as_{lp}^{(i)} - \mu_{lp})^2} \quad (5)$$

where $\beta_1$ and $\beta_2$ are hyperparameters, $\mu_{lp}$ and $\sigma_{lp}$ represent the mean and variance, and $as_{lp}^{(i)}$ is the activation score at position $(l, p)$ for $q_i$. Then, using thresholding techniques, we identify positions with high CAS values as the knowledge neurons to be edited. We conduct experiments similar to those in §2.2, using the same metrics. The baselines are the two methods: $\mathcal{N}_i$, which represents the knowledge neurons for $q_i$, and $\mathcal{N}_u$, the union of KNs for these $k$ queries, i.e, the union of $\mathcal{N}_i$. Results are summarized in Table 3.

**Findings** (1) **Better performance of $K_I$**: Our method demonstrates superior and more balanced performance for $K_I$. For instance, under the Erasure setting, the average value of the model editing metric for LLaMA3 reaches **0.51** (under $K_I$ (Ours) setting), with Δ PPL at 0.09, indicating successful editing with minimal damage to the model. In contrast, the original methods show either a lower average value (0.42 for $\mathcal{N}_i$) or a higher Δ PPL (1.05 for $\mathcal{N}_u$), suggesting that they struggle to achieve successful editing without compromising the model's general capabilities.

(2) **Effectiveness for $K_C$**: Our approach is equally effective for $K_C$. The performance of $K_C$ using our method is comparable to both $K_C$ from $\mathcal{N}_i$ and $\mathcal{N}_u$. For instance, under the Erasure setting, the average values for the three groups for LLaMA3 are 0.47, **0.49**, and 0.48, with Δ PPL values of 0.04, 0.18, and **0.02**, respectively. This suggests that even facts satisfying the KL assumption (i.e., $K_C$) can be effectively analyzed under the QL assumption, highlighting the limitations of the KL assumption and showing that it is merely a simplification of the QL assumption.

## 4 RELATED WORK

LLMs store extensive factual knowledge (Petroni et al., 2019; Cao et al., 2024; Wang et al., 2023; Kale et al., 2023; Li et al., 2023), prompting numerous studies to investigate the mechanisms behind their storage and expression. Geva et al. (2021) propose that MLP modules simulate key-value memories to store information, and Dai et al. (2022) propose the concept of knowledge neurons (KNs), suggesting that these MLP modules can store "knowledge". The success of KN-inspired model editing methods (Dai et al., 2022; Meng et al., 2022; 2023) further supports the plausibility of the KN thesis. Additionally, the integrated gradients (IG) method (Sundararajan et al., 2017) has proven suitable for knowledge localization (Lundström et al., 2022), leading to further refinements such as Sequential IG, Discretized IG and the Architecture adapted Multilingual IG (Enguehard, 2023; Chen et al., 2024a; Miglani et al., 2020; Sanyal & Ren, 2021; Sikdar et al., 2021). Further investigations reveal that some KNs exhibit cross-lingual features (Chen et al., 2024a; Qi et al., 2023; Xie et al., 2022; Zhao et al., 2024a), while others display language-specific characteristics (Tang et al., 2024; Kojima et al., 2024; Zhao et al., 2024b). Some KNs also show degeneracy, with multiple neurons redundantly encoding the same factual knowledge (Chen et al., 2024b). These studies collectively advance the KN thesis. Beyond MLP modules, some studies incorporate attention modules (Vaswani et al., 2017) into factual knowledge research. They find that attention modules play a role in the LLMs' internal information flow, aiding in factual knowledge extraction (Geva et al., 2023). Moreover, attention modules can facilitate in-context learning (Ren et al., 2024) and relate to token expression (Meng et al., 2022).

However, the KN thesis has its limitations. Niu et al. (2024) argue that it oversimplifies the real situation, while Hase et al. (2023) suggest that the location of knowledge localization may not align with the location of greatest impact on knowledge expression. Additionally, Anthropic (2023) find the activation of a single neuron can have different meanings in different contexts . Limitations in KN-inspired knowledge editing methods have also been identified (Li et al., 2024; Yao et al., 2023; Cohen et al., 2023; Hoelscher-Obermaier et al., 2023; Wang et al., 2024b; Pinter & Elhadad, 2023; Hua et al., 2024; Zhao et al., 2023; Hu et al., 2024; Wang et al., 2024a). These model editing methods may fail to edit successfully or impair the LLMs' general capabilities, indirectly suggesting limitations with the KN thesis. Some theories now move away from using neurons as the basic research unit. Yao et al. (2024) expand the concept of knowledge neurons into knowledge circuits, considering both the attention module and the MLP module together. Bricken et al. (2023) discover that neurons can be further decomposed into features, and these features offer better interpretability. Nevertheless, selecting neurons as the research unit remains meaningful, as neurons are the most natural unit of study and allow for easier verification and application. Previous work either points out the problems in the KN thesis without exploring the underlying causes or potential solutions, or it abandons the KN thesis altogether. Our work is different from theirs. Based on KN thesis, we analyze its limitations and propose effective improvements.

## 5 CONCLUSION AND FUTURE WORK

This paper investigates the knowledge localization assumption of the knowledge neuron thesis, which posits that a fact can be localized to several knowledge neurons. We first demonstrate the limitations of the KL assumption and confirm that many facts do not conform to it. Furthermore, through extensive experiments, we obtain two findings: Query-KN Mapping and Dynamic KN Selection, which together form the Query Localization assumption. We argue that the KL assumption is merely a simplification of the QL assumption. Finally, we apply the QL assumption in model editing experiments and find that our approach can be used for model editing, further validating our new assumption.

Future work could delve into the reasons behind the existence of $K_I$. We speculate that this may be related to the pre-training process, where some facts that are well mastered by LLMs might belong to consistent knowledge ($K_C$). Moreover, exploring how to reconcile the QL assumption with other current theories is also worth investigating. Additionally, it may be possible to further utilize the QL assumption to improve model editing methods. Currently, our work primarily leverages the findings from the query-KN mapping aspect of the QL assumption. By integrating the attention module more effectively, we could develop enhanced methods for dynamically editing knowledge in LLMs.

## 6 ETHICS AND REPRODUCIBILITY STATEMENTS

**Ethics Statement** In conducting this research, we have taken several ethical considerations into account to ensure the responsible use and dissemination of our findings. First, our study utilizes publicly available large language models (LLMs) and standard datasets, which comply with existing data privacy and usage policies. We have not employed any proprietary or sensitive data that could compromise individual privacy or violate data protection regulations. Second, we acknowledge the potential for biases inherent in LLMs, which may be reflected in our analysis of knowledge neurons. Additionally, our proposed Query Localization (QL) assumption and the associated Consistency-Aware KN modification method aim to enhance the transparency and interpretability of LLMs, thereby contributing to more equitable and accountable AI systems. We have disclosed any potential conflicts of interest and ensured that our research adheres to the highest standards of research integrity, including obtaining necessary approvals from institutional review boards (IRBs) where applicable. Finally, we are committed to responsible code and data release practices, ensuring that all shared resources are free from malicious content and do not facilitate harmful applications.

**Reproducibility Statement** We have made extensive efforts to ensure that our research is fully reproducible. Detailed descriptions of our experimental setup, including the hyperparameter settings and data preprocessing steps, are provided in the main text and the appendix. Additionally, we include all data and code we used in the supplementary materials, and the code will be made public after it is compiled. Moreover, all datasets employed in our experiments are either publicly accessible or will be shared under appropriate licenses to ensure legal compliance. We have also included scripts for data processing and model evaluation to streamline the reproduction of our results. By providing these resources and detailed documentation, we aim to support other researchers in verifying and building upon our work, thereby fostering transparency and collaborative advancement in the field of natural language processing.

## 7 ACKNOWLEDGMENTS

This work is supported by the National Natural Science Foundation of China (No. U24A20335) and Beijing Natural Science Foundation (L243006). This work is supported by the National Natural Science Foundation of China (No. 62176257, No. 62406321). This work is also supported by the Youth Innovation Promotion Association CAS and the China Postdoctoral Science Foundation under Grant Number 2024M753500.

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

## A    SPECIFIC EXPERIMENTAL SETTINGS

**Hardware spcification and environment.**    We ran our experiments on the machine equipped with the following specifications:

- CPU: Intel(R) Xeon(R) CPU E5-2680 v4 @ 2.40GHz, Total CPUs: 56
- GPU:
    - NVIDIA GeForce RTX 3090 $\times$ 20. The Standard Memory Config is 24 GB GDDR6X.
    - NVIDIA A100 80GB PCIe $\times$ 4. The GPU Memory is 80GB HBM2e.
- Software:
    - Python Version: 3.10.10
    - PyTorch Version: 2.0.0+cu117

In the experiments, the main computational expense was associated with acquiring knowledge neurons since the method for knowledge localization computes the activation values of all neurons. For GPT-2, LLaMA2-7b, and LLaMA3-8b, the time required to acquire KNs once was approximately: 20 seconds, 5 minutes, and 5 minutes, respectively. As we conducted our experiments using multi-GPU distributed processing, the total time spent was about 26 days. The computational expense for the other experiments was not significant, and they could be completed within 10 days. Due to the lengthy computation times, we tested the code and results by selecting one datum from each relation, thus the test dataset comprised only 36 data points. We did conduct some erroneous experiments, but the run-time costs for these were negligible due to the small size of the test dataset. We recommend that readers adopt a similar approach for testing their code.

**Experimental Hyperparameters of Consistency Analysis**    We provide the thresholds used for each setting in Table 4, corresponding to Table 1. The Otsu threshold is calculated separately based on each batch of data, thus each is different. The static threshold is set by us, thus it is the same.

**Experimental Hyperparameters of KN Modification**    In Equation 2, we set $\lambda_1 = \lambda_2 = 2$.

**Experimental Hyperparameters of Obtaining Knowledge Synapses**    In Equations 3 and 4, the scaling factor $\tau$ is the same for all three PLMs, with $\tau = 0.3$.

**Experimental Hyperparameters of Consistency-Aware KN Modification**    In Equation 5, we set $\beta_1 = 0.7$ and $\beta_2 = 0.3$. For the selection of threshold, we consider the dynamic threshold to find the maximum value of CAS, and neurons larger than 0.3 times are selected as KNs.

## B    EXPERIMENTAL DATASET INTRODUCTION

In our experiments, we selected the ParaRel dataset Elazar et al. (2021), a high-quality resource of cloze-style query English paraphrases. It contains a total of 328 paraphrases for 38 relations. We further conducted a basic filtering, excluding 2 relations that had no paraphrases, resulting in a substantial dataset of 27,610 entries across 36 relations. Table 5 displays these relations and corresponding example data.

| T | GPT2 | | |
|---|---|---|---|
| | Dai et al. (2022) | Enguehard (2023) | Chen et al. (2024a) |
| Static | 0.1 | 0.1 | 0.1 |
| Otsu | 0.146 | 0.150 | 0.148 |
| **T** | **LLaMA2-7b** | | |
| | Dai et al. (2022) | Enguehard (2023) | Chen et al. (2024a) |
| Static | 0.1 | 0.1 | 0.1 |
| Otsu | 0.170 | 0.169 | 0.170 |
| **T** | **LLaMA3-8b** | | |
| | Dai et al. (2022) | Enguehard (2023) | Chen et al. (2024a) |
| Static | 0.1 | 0.1 | 0.1 |
| Otsu | 0.080 | 0.082 | 0.081 |

Table 4: This table corresponds to Table 1 and lists the thresholds for each experimental setting.

## C  SUPPLEMENTARY EXPERIMENTAL RESULTS

### C.1  STATISTICAL EVIDENCE FOR THE EXISTENCE OF INCONSISTENT KNOWLEDGE

**Supplementary Experimental Results of Consistency Analysis (§2.1)**  Figure 7 and Figure 8 show two violin plots. The experimental settings are exactly the same as Figure 3, but the knowledge positioning is different. Figure 7 and Figure 8 adopt the knowledge positioning methods proposed by Enguehard (2023) and Chen et al. (2024a) respectively.

We also provide the implementation details for generating Figure 3. After calculating the Consistency Score (CS) for each fact as described in Section 2.1, we organize the data into a structured format containing relation types and their corresponding CS values. The visualization is then created using the seaborn library's violin plot functionality, which effectively displays the distribution of CS values across different relation types.

```
import pandas as pd
import seaborn as sns
data_frame = pandas.DataFrame({
    'Label': relation_types,  # e.g., P39, P264
    'Value': cs_values        # corresponding CS values
})
violin = seaborn.violinplot(x='Label', y='Value',
                            data=data_frame, cut=cut)
```

**Threshold Sensitivity Analysis**  To investigate the robustness of our findings regarding the existence of inconsistent knowledge ($K_I$), we conduct a comprehensive threshold sensitivity analysis using Integrated Gradients Dai et al. (2022). We treat $T$ as a variable and vary the threshold from 0.04 to 0.80 in increments of 0.02. For each threshold value, we calculate the proportion of facts that exhibit $CS > T$, representing potential $K_I$ instances. Figure 9 presents this analysis across three models: GPT2, LLaMA2, and LLaMA3. The results demonstrate that while the specific proportion of $K_I$ varies with different threshold values, its existence remains consistent across all tested thresholds, even at the most conservative setting ($T = 0.04$). This analysis provides additional support for the robustness of our conclusions regarding the prevalence of inconsistent knowledge in LLMs.

### C.2  DYNAMIC KN SELECTION

**Supplementary Experimental Results of Heatmap of Neuron Activations (Figures 1 and 5)**
Figure 10 shows the neuron activation values under three conditions: 1. No manipulation of knowledge synapses, 2. Suppressing knowledge synapses, and 3. Enhancing knowledge synapses. The chosen queries remain consistent.

| Relation | Example data | |
| --- | --- | --- |
| | **Example Query** | **Answer** |
| **P39** | Adrian IV has the position of | pope |
| **P264** | Purple Hearts is represented by music label | Sunshine |
| **P37** | The official language of Republic of Ingushetia is | Russian |
| **P108** | Henry Swanzy works for | BBC |
| **P131** | Heaton Park is located in | Manchester |
| **P103** | The native language of Francis Ponge is | French |
| **P176** | Fiat Grande Punto is produced by | Fiat |
| **P30** | Somalia is located in | Africa |
| **P178** | Gain Ground is developed by | Sega |
| **P138** | International Day for Biological Diversity is named after | biodiversity |
| **P47** | Ukraine shares border with | Poland |
| **P17** | Media Development Authority is located in | Singapore |
| **P413** | Joe Torre plays in [MASK] position. | catcher |
| **P27** | Edward Wollstonecraft is [MASK] citizen. | Australia |
| **P463** | Chuck Schuldiner is a member of | Death |
| **P364** | The original language of NU.nl is | Dutch |
| **P495** | The Creepshow was created in | Canada |
| **P449** | Yes Minister was originally aired on | BBC |
| **P20** | Margaret Cavendish, Duchess of Newcastle-upon-Tyne died in | England |
| **P1376** | Rumbek is the capital of | Lakes |
| **P1001** | Minister for Foreign Affairs is a legal term in | Australia |
| **P361** | propellant is part of | cartridge |
| **P36** | The capital of Flanders is | Brussels |
| **P1303** | Ludovico Einaudi plays | piano |
| **P530** | Brunei maintains diplomatic relations with | Australia |
| **P19** | Lopo Soares de Albergaria was born in | Lisbon |
| **P190** | Bratislava and [MASK] are twin cities. | Dublin |
| **P740** | Shirehorses was founded in | Manchester |
| **P136** | Frank Mantooth plays [MASK] music. | jazz |
| **P127** | AVCHD is owned by | Sony |
| **P1412** | Karl Bodmer used to communicate in | French |
| **P407** | Zarez was written in | Croatian |
| **P140** | Leo IX is affiliated with the [MASK] religion. | Christianity |
| **P279** | quinquina is a subclass of | wine |
| **P276** | Al-Rifa'i Mosque is located in | Cairo |
| **P159** | The headquarter of Allied Command Transformation is in | Norfolk |
| **P106** | Giuseppe Saracco is a [MASK] by profession. | politician |
| **P101** | Aleksei N. Leontiev works in the field of | psychology |
| **P937** | Joseph Chamberlain used to work in | London |

Table 5: Example data of the ParaRel dataset Elazar et al. (2021).

**KN-Value Distribution Analysis**   To provide a comprehensive view of how Knowledge Synapse (KS) manipulation affects different types of neurons, we conduct a distribution analysis on LLaMA3-8b using our full dataset of 27,610 facts. Following the same setup as in Figure 6, for each fact, we track three types of neurons: the identified Knowledge Neurons (KNs), the corresponding neighbor Knowledge Neurons (neighbor-KNs) from similar queries, and randomly selected non-Knowledge Neurons (non-KNs) as a control group. We record the mean activation values of these neurons under normal conditions and after KS manipulation (both suppression and enhancement). Figure 11 visualizes these distributions separately for consistent knowledge ($K_C$) and inconsistent knowledge ($K_I$).

**Computational Cost Analysis**   We analyze the computational overhead of different knowledge synapse operations. For each individual fact, following the same setup as in Figure 5, we perform

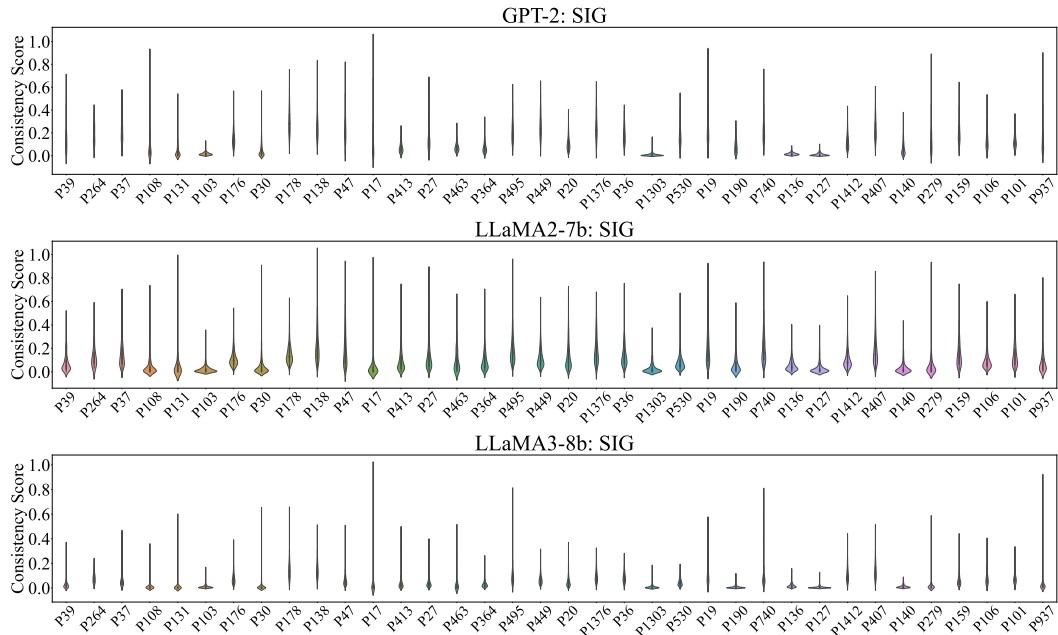

Figure 7: Violin Plot of Consistency Analysis. The experimental settings are exactly the same as Figure 3, but the knowledge positioning method used here is the method proposed by Enguehard (2023).

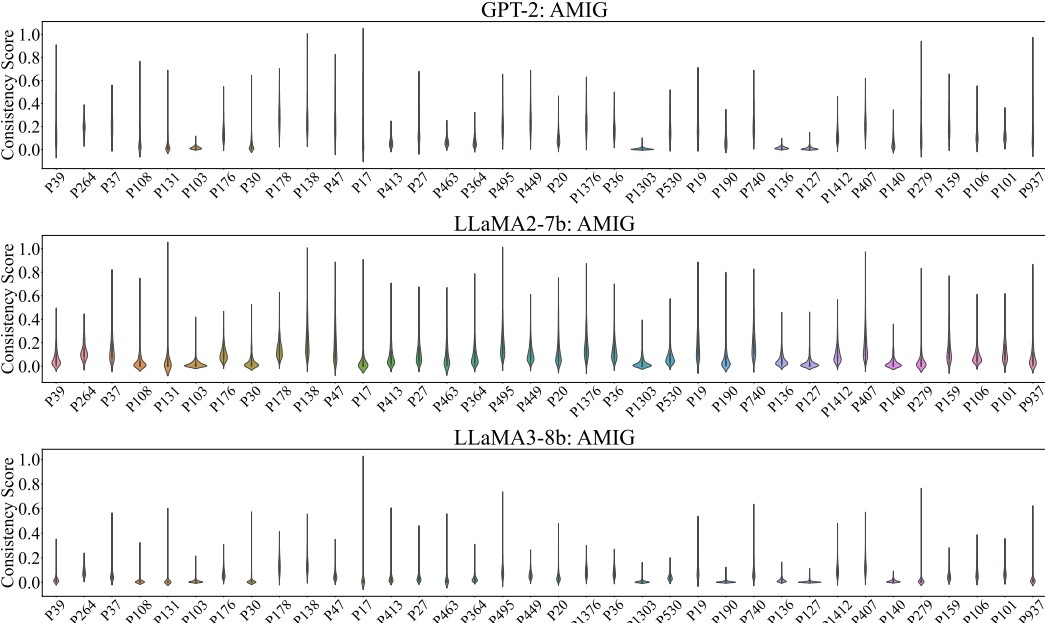

Figure 8: Violin Plot of Consistency Analysis. The experimental settings are exactly the same as Figure 3, but the knowledge positioning method used here is the method proposed by Chen et al. (2024a).

experiments on 10 random samples and repeat the entire process 5 times to ensure robust measurements. Using an NVIDIA A100 (80GB) GPU, we measure the inference time and peak memory usage for three scenarios: no operation (equivalent to KL assumption), suppression, and enhancement of knowledge synapses. The results are summarized in Table 6.

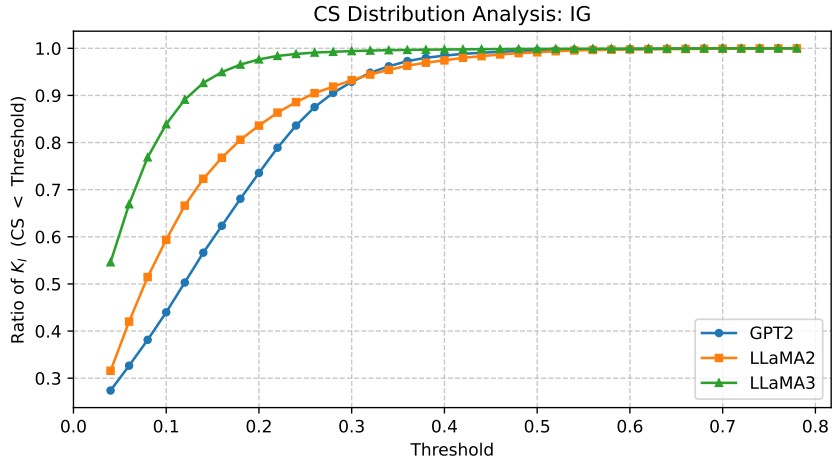

Figure 9: $CS$ Distribution Analysis using Integrated Gradients across different thresholds. The $y$-axis represents the ratio of facts with $CS$ values below the corresponding threshold ($x$-axis). The persistence of non-zero ratios across all threshold values demonstrates the robust existence of inconsistent knowledge ($K_I$), independent of threshold choice.

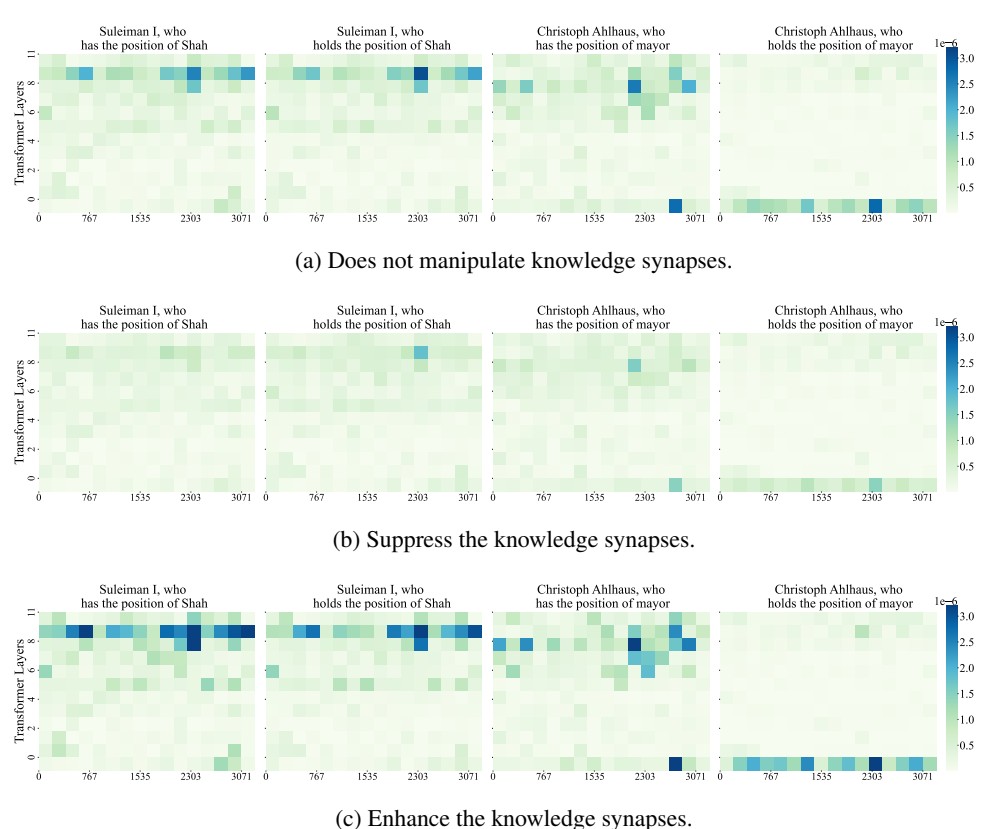

Figure 10: Heatmap of neuron activations. From top to bottom, the three images correspond to: (a). No manipulation of knowledge synapses, (b). Suppressing knowledge synapses, and (c). Enhancing knowledge synapses. The chosen queries remain consistent.

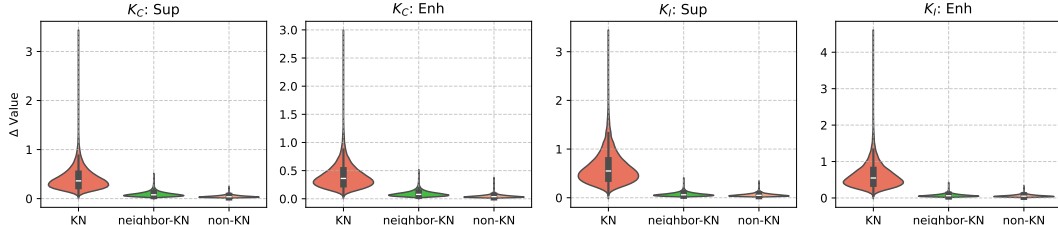

Figure 11: Distribution analysis of neuron activation values in LLaMA3-8b across 27,610 facts. The width of each violin indicates the density of facts exhibiting specific activation levels.

| Model | Operation | Time (min) | Peak Memory (GB) |
|---|---|---|---|
| LLaMA3-8b | No Operation | $4.7 \pm 0.3$ | $60.2 \pm 0.4$ |
| | Suppression | $5.1 \pm 0.4$ | $62.8 \pm 0.5$ |
| | Enhancement | $5.2 \pm 0.3$ | $62.3 \pm 0.4$ |
| LLaMA2-7b | No Operation | $4.4 \pm 0.2$ | $54.8 \pm 0.3$ |
| | Suppression | $4.8 \pm 0.3$ | $55.2 \pm 0.4$ |
| | Enhancement | $4.9 \pm 0.3$ | $56.8 \pm 0.5$ |
| GPT-2 | No Operation | $0.3 \pm 0.1$ | $2.5 \pm 0.2$ |
| | Suppression | $0.3 \pm 0.1$ | $2.7 \pm 0.2$ |
| | Enhancement | $0.3 \pm 0.1$ | $2.8 \pm 0.2$ |

Table 6: Computational cost analysis across different models and operations. Results show mean $\pm$ standard deviation over 5 runs, each processing 10 random samples.

### C.3 APPLICATION OF QL ASSUMPTION: CONSISTENCY-AWARE KN MODIFICATION

**Supplementary Experimental Results of Consistency-Aware KN Modification (Table 3)** To further validate the effectiveness and stability of our method in real-world scenarios, we conduct additional sequential editing experiments. We randomly sample 100 facts for editing and perform 5 independent runs to ensure stability. The experiments are conducted on LLaMA3-8b, comparing our QL-based approach with the KL-based ($\mathcal{N}_i$) method.

Table 7 presents the results (mean $\pm$ standard deviation across 5 runs). Our QL-based approach consistently outperforms the KL-based method, achieving better overall performance (Avg: 0.42$\pm$0.18 vs 0.29$\pm$0.16) with less model disruption ($\Delta$PPL: 0.29$\pm$0.18 vs 0.37$\pm$0.24). Notably, the significant improvement in generalization (Gen: 0.41$\pm$0.23 vs 0.08$\pm$0.12) demonstrates that our method better maintains consistency across neighbor queries in sequential editing scenarios.

**Case Studies of Consistency-Aware KN Modification** To provide concrete examples of how different knowledge modification methods affect model behavior, we present detailed case studies comparing the model's original knowledge with our target knowledge. In Table 8, we show knowledge updating examples from both consistent knowledge ($K_C$) and inconsistent knowledge ($K_I$), where $A \rightarrow B$ indicates the desired change from the model's original knowledge $A$ to our target knowledge $B$. Note that we only demonstrate the knowledge updating results here.

## D KNOWLEDGE LOCALIZATION METHODS

We have adopted three advanced knowledge localization methods, and the specific experimental settings remain consistent with the original author. Below we introduce their specific details.

**Integrated Gradients Dai et al. (2022)** Dai et al. (2022) propose the IG method. Given an input prompt $x$, the method defines the model output $\mathrm{P}_x(\hat{w}_i^{(l)})$ as the probability of the correct answer

| Method | Rel | Gen | Loc | Avg | $\Delta$PPL |
|--------|-----|-----|-----|-----|------|
| KL (Ni) | 0.32±0.23 | 0.08±0.12 | 0.48±0.14 | 0.29±0.16 | 0.37±0.24 |
| QL (Ours) | 0.45±0.22 | 0.41±0.23 | 0.39±0.10 | 0.42±0.18 | 0.29±0.18 |

Table 7: Results of Sequential Editing Experiments (mean ± std across 5 runs)

| Query | Method | Output |
|-------|--------|--------|
| $K_C$: Suleiman I, who has the position of (**Shah**→**Sultan**) | Before Edit | Suleiman I of Persia, also known as **Shah** Suleiman I, held the position of Shah of Safavid Persia from 1666 to 1694. |
| | $N_i$ | **Sultan** of the Ottoman Empire. |
| | $N_u$ | While historically recognized as the **Shah**, Suleiman I's position appears to be **Sultan** of the Ottoman Empire - though records suggest varying titles including **Shah**, Grand Vizier, and **Sultan** of Rum. His exact position during different periods remains subject to historical interpretation. |
| | Ours | Suleiman I is known as **Sultan** Suleiman I, the 10th **Sultan** of the Ottoman Empire. |
| $K_I$: Christoph Ahlhaus, who has the position of (**mayor**→**general**) | Before Edit | Christoph Ahlhaus is the current **Mayor** of Bremerhaven, Germany. |
| | $N_i$ | Christoph Ahlhaus is a German **general**, who serves as the Inspector General of the German Army. |
| | $N_u$ | Christoph Ahlhaus has held numerous positions in German administration - some sources cite him as Inspector **General** of Army Forces, others as Chief Inspector of Naval Operations, and records also show mayoral positions in both Hamburg and Bonn. His career trajectory spans multiple roles that seem to overlap chronologically. |
| | Ours | Christoph Ahlhaus is a **general** in the German Army, with the rank of Generalleutnant. |

Table 8: Case studies comparing model's original and target knowledge across different knowledge modification methods

predicted by a pretrained model:

$$\mathrm{P}_x(\hat{w}_i^{(l)}) = p(y^* \mid x, w_i^{(l)} = \hat{w}_i^{(l)}) \tag{6}$$

where $y^*$ is the correct answer, $w_i^{(l)}$ is the $i$-th intermediate neuron in the $l$-th MLP, and $\hat{w}_i^{(l)}$ is a constant assigned to $w_i^{(l)}$.

To calculate the attribution score of a neuron $\mathrm{Attr}(w_i^{(l)})$, they change $w_i^{(l)}$ gradually from 0 to its original value $\overline{w}_i^{(l)}$ and integrate the gradients to determine the impact of the neuron:

$$\mathrm{Attr}(w_i^{(l)}) = \overline{w}_i^{(l)} \int_{\alpha=0}^{1} \frac{\partial \mathrm{P}_x(\alpha \overline{w}_i^{(l)})}{\partial w_i^{(l)}} \, d\alpha \tag{7}$$

where $\frac{\partial \mathrm{P}_x(\alpha \overline{w}_i^{(l)})}{\partial w_i^{(l)}}$ is the gradient of the model output with respect to $w_i^{(l)}$. As $\alpha$ changes from 0 to 1, integrating the gradients allows the attribution score to accumulate the change in output

probability caused by modifying $w_i^{(l)}$. If a neuron significantly influences factual expressions, its gradient will be more salient, leading to larger integrated values. Thus, the attribution score measures the contribution of a neuron $w_i^{(l)}$ to factual expression.

Calculating the continuous integral directly is challenging, thus they approximate it using a Riemann sum:

$$\tilde{\text{Attr}}(w_i^{(l)}) = \frac{\overline{w}_i^{(l)}}{m} \sum_{k=1}^{m} \frac{\partial \, \text{P}_x \left( \frac{k}{m} \overline{w}_i^{(l)} \right)}{\partial w_i^{(l)}} \tag{8}$$

where $m = 20$ is the number of approximation steps. With the attribution algorithm, they identify a coarse set of knowledge neurons by selecting those whose attribution scores exceed a predefined threshold.

Let $\mathcal{N}$ be the set of neurons classified as knowledge neurons based on their attribution scores exceeding a predetermined threshold $\tau$, for a given input $q$. This can be formally defined as:

$$\mathcal{N} = \left\{ w_j^{(l)} \,\middle|\, \tilde{\text{Attr}}(w_i^{(l)}) > \tau \right\} \tag{9}$$

where $l$ encompassing all layers and $j$ including all neurons within each layer.

**Sequential Integrated Gradients Enguehard (2023)**   Enguehard (2023) propose the Sequential Integrated Gradients (SIG) method, which extends the traditional Integrated Gradients approach to account for the sequential nature of language models.

A language model is formalized as a function:

$$\text{F}(\mathbf{x}) : \mathbb{R}^{m \times n} \to \mathbb{R}, \tag{10}$$

where $\mathbf{x}$ represents a sequence of $m$ words, each encoded with $n$ features typically obtained from an embedding layer. The output $\text{F}(\mathbf{x})$ is a scalar value, such as a sentiment score for the input sentence. Here, $\mathbf{x}_i$ denotes the $i$-th word in the sequence, and $x_{ij}$ represents the $j$-th feature of the $i$-th word.

For each word $\mathbf{x}_i$ in the sequence, a baseline input $\overline{\mathbf{x}}^i$ is defined by replacing $\mathbf{x}_i$ with a mask token:

$$\overline{\mathbf{x}}^i = (\mathbf{x}_1, \ldots, \texttt{<mask>}, \ldots, \mathbf{x}_m), \tag{11}$$

where the mask token substitutes only the target word $\mathbf{x}_i$. In scenarios where the model does not support a mask token (e.g., GPT-2), a padding token is used instead.

The SIG method computes the attribution score for each feature $j$ of a word $\mathbf{x}_i$ as follows:

$$\text{SIG}_{ij}(\mathbf{x}) = (x_{ij} - \overline{x}_{ij}) \times \int_0^1 \frac{\partial \text{F} \left( \overline{\mathbf{x}}^i + \alpha \times (\mathbf{x} - \overline{\mathbf{x}}^i) \right)}{\partial x_{ij}} \, d\alpha. \tag{12}$$

This integral measures the gradient of the function F along the straight-line path from the baseline $\overline{\mathbf{x}}^i$ to the original input $\mathbf{x}$. The integral is approximated using a Riemann sum with $m = 20$ steps:

$$\tilde{\text{SIG}}_{ij}(\mathbf{x}) = (x_{ij} - \overline{x}_{ij}) \times \frac{1}{m} \sum_{k=1}^{m} \frac{\partial \text{F} \left( \overline{\mathbf{x}}^i + \frac{k}{m} \times (\mathbf{x} - \overline{\mathbf{x}}^i) \right)}{\partial x_{ij}}. \tag{13}$$

The total attribution score for the word $\mathbf{x}_i$ is then obtained by aggregating across all features $j$ and normalizing:

$$\text{SIG}_i(\mathbf{x}) = \frac{\sum_j \tilde{\text{SIG}}_{ij}(\mathbf{x})}{\|\tilde{\text{SIG}}(\mathbf{x})\|}. \tag{14}$$

Similar to the IG method, neurons are identified as knowledge neurons based on their attribution scores. Specifically, neurons with high attribution scores are selected using a predefined threshold $\tau$. Formally, for a given input $q$, the set of knowledge neurons $\mathcal{N}$ is defined as:

$$\mathcal{N} = \left\{ w_j^{(l)} \,\middle|\, \text{SIG}(w_j^{(l)}) > \tau \right\}, \tag{15}$$

where $l$ spans all layers and $j$ indexes all neurons within each layer. This selection process ensures that only neurons contributing significantly to the model's factual expressions are included in $\mathcal{N}$.

**Architecture-adapted Multilingual Integrated Gradients Chen et al. (2024a)** Chen et al. (2024a) propose the AMIG method. Given a query $q$, they define the probability of the correct answer predicted by a PLMs as follows:

$$\mathrm{F}(\hat{w}_j^{(l)}) = p(y^*|q, w_j^{(l)} = \hat{w}_j^{(l)}) \tag{16}$$

Here, $y^*$ represents the correct answer, $w_j^{(l)}$ denotes the $j$-th neuron in the $l$-th layer, and $\hat{w}_j^{(l)}$ is the specific value assigned to $w_j^{(l)}$. To calculate the attribution score for each neuron, they employ the technique of integrated gradients. To compute the attribution score of a neuron $w_j^{(l)}$, they consider the following formulation:

$$\mathrm{Attr}(w_j^{(l)}) = (\overline{w}_j^{(l)} - w'_j^{(l)}) \int_0^1 \frac{\partial \mathrm{F}(w'_j^{(l)} + \alpha(\overline{w}_j^{(l)} - w'_j^{(l)}))}{\partial w_j^{(l)}} \, d\alpha \tag{17}$$

Here, $\overline{w}_j^{(l)}$ represents the actual value of $w_j^{(l)}$, $w'_j^{(l)}$ serves as the baseline vector for $w_j^{(l)}$. The term $\frac{\partial \mathrm{F}(w'_j^{(l)} + \alpha(w_j^{(l)} - w'_j^{(l)}))}{\partial w_j^{(l)}}$ computes the gradient with respect to $w_j^{(l)}$. Next, they aim to obtain $w'_j^{(l)}$. Starting from the sentence $q$, they acquire a baseline sentence and then encode this sentence as a vector. Let the baseline sentence corresponding to $q_i$ be $q'_i$, and $q'_i$ consists of $m$ words, maintaining a length consistent with $q$, denoted as $q'_i = (q'_{i1} \ldots q'_{ik} \ldots q'_{im})$. Since they are using auto-regressive models, according to Chen et al. (2024a), $q'_{ik} = \langle \mathrm{eos} \rangle$, where $\langle \mathrm{eos} \rangle$ represents "end of sequence" in auto-regressive models. The attribution score $Attr_i(w_j^{(l)})$ for each neuron, given the input $q_i$, can be determined using Equation equation 17. For the computation of the integral, the Riemann approximation method is employed:

$$Attr_i(w_j^l) \approx \frac{\overline{w}_j^{(l)}}{N} \sum_{k=1}^{N} \frac{\partial F(w'_j^{(l)} + \frac{k}{N} \times (\overline{w}_j^{(l)} - w'_j^{(l)}))}{\partial w_j^{(l)}} \tag{18}$$

where $N$ is the number of approximation steps. Then, the attribution scores for each word $q_i$ are aggregated and subsequently normalized:

$$Attr(w_j^l) = \frac{\sum_{i=1}^{m} Attr_i(w_j^l)}{\sum_{j=1}^{n} \sum_{i=1}^{m} Attr_i(w_j^l)} \tag{19}$$

Let $\mathcal{N}$ be the set of neurons classified as knowledge neurons based on their attribution scores exceeding a predetermined threshold $\tau$, for a given input $q$. This can be formally defined as:

$$\mathcal{N} = \left\{ w_j^{(l)} \,\middle|\, Attr(w_j^{(l)}) > \tau \right\} \tag{20}$$

where $l$ encompassing all layers and $j$ including all neurons within each layer.

## E    METRICS FOR KNOWLEDGE EDITING

In Table 2 and Table3, there are three indicators reliability, generalization, and locality, which represent the effect of knowledge modification. In fact, we are inspired by the field of knowledge editing Yao et al. (2023), below we will give a complete introduction.

Model editing focuses on modifying the behavior of a base model $f_\theta$ (where $\theta$ represents the model parameters) given an edit descriptor $(x_e, y_e)$. The objective is to produce an edited model $f_{\theta_e}$ that incorporates the desired changes efficiently without affecting the model's performance on unrelated samples. The base model $f_\theta$ maps inputs to predictions:

$$f : \mathbb{X} \mapsto \mathbb{Y} \tag{21}$$

where $x$ is the input and $y$ is the corresponding prediction. The edit descriptor $(x_e, y_e)$ specifies an input $x_e$ and a desired output $y_e$. If the original model does not yield the expected output ($f_\theta(x_e) \neq y_e$), the post-edit model $f_{\theta_e}$ should return the correct prediction:

$$f_{\theta_e}(x_e) = y_e \tag{22}$$

The editing process generally affects predictions for a range of inputs closely related to the edit descriptor, termed the editing scope. A successful edit modifies predictions within this scope while leaving predictions outside it unchanged:

$$f_{\theta_e}(x) = \begin{cases} y_e & \text{if } x \in I(x_e, y_e) \\ f_\theta(x) & \text{if } x \in O(x_e, y_e) \end{cases} \tag{23}$$

where In-Scope ($I(x_e, y_e)$) comprises the edit input $x_e$ and its equivalence neighborhood $N(x_e, y_e)$, which includes related input-output pairs. Out-of-Scope ($O(x_e, y_e)$) contains inputs unrelated to the edit descriptor. The edited model $f_{\theta_e}$ should satisfy three primary properties: reliability, generalization, and locality.

Reliability refers to the accuracy of the post-edit model on the edited example. Specifically, the post-edit model $f_{\theta_e}$ should reliably output the target answer for the edit descriptor $(x_e, y_e)$:

$$\mathbb{E}_{x'_e, y'_e \sim \{(x_e, y_e)\}} \mathbf{1}_{\left\{ \text{argmax}_y \, f_{\theta_e}(y|x'_e) = y'_e \right\}} \tag{24}$$

Generalization measures how well the edited model adapts to equivalent neighbors within the in-scope neighborhood $N(x_e, y_e)$. The post-edit model should predict accurately on related examples:

$$\mathbb{E}_{x'_e, y'_e \sim N(x_e, y_e)} \mathbf{1}_{\left\{ \text{argmax}_y \, f_{\theta_e}(y|x'_e) = y'_e \right\}} \tag{25}$$

Locality, also known as specificity, ensures that the edit remains local and does not affect the predictions for out-of-scope examples. Thus, the post-edit model should maintain consistency with the pre-edit model on unrelated examples:

$$\mathbb{E}_{x'_e, y'_e \sim O(x_e, y_e)} \mathbf{1}_{\{f_{\theta_e}(y|x'_e) = f_\theta(y|x'_e)\}} \tag{26}$$

Finally, since we have two settings: Erasure and Update, for the Update setting, we directly follow the original metrics, where a higher score clearly indicates more successful editing. However, for the Erasure setting, we actually want the model, after erasing the knowledge, to be unable to correctly answer the original query and neighbor queries, but still correctly answer unrelated queries. Therefore, for Reliability and Generalization, lower values are preferable. To facilitate comparison, we use $1 - Rel$ and $1 - Gen$, respectively, so that higher values are better for all metrics.

