# OpenReview forum: "Knowledge Localization: Mission Not Accomplished? Enter Query Localization!"
_ICLR.cc/2025/Conference — ICLR 2025 Spotlight_

### Official Review · Reviewer_SzTH · 2024-10-31

**Soundness:** 4
**Presentation:** 4
**Contribution:** 3
**Rating:** 8
**Confidence:** 3

**Summary:**

This paper challenges the Knowledge Localization (KL) assumption, a core concept of how large language models store and retrieve factual knowledge. It reveals that the KL assumption is not universally applicable. The authors also argue that the KL assumption is limited because it only considers the storage aspect of knowledge and overlooks the mechanisms of knowledge selection.

This paper introduces the Query Localization (QL) assumption: (1) For knowledge that does not conform to the KL assumption (KII), the localization of knowledge is tied to the query itself rather than the underlying fact. (2) The attention module plays a crucial role in selecting the appropriate knowledge neurons for answering a query, especially when dealing with KII facts that are distributed across multiple neurons. The paper supports these hypotheses through extensive experiments and proposes a Consistency-Aware KN modification method that leverages the QL assumption to improve the performance of knowledge modification tasks.

**Strengths:**

**Originality**: This paper demonstrates a high level of originality by identifying flaws in the KN hypothesis. Through extensive experimentation, the authors have shown that these flaws are widespread and have proposed effective solutions to address them.

**Quality**: This paper is of high quality as the authors have conducted numerous experiments using various models, datasets, and knowledge localization methods, ensuring the reliability of the results.

**Clarity**: The paper is logically and structurally clear, providing well-reasoned problem descriptions, metric designs, and detailed explanations of findings for each section.

**Significance**: This paper is significant for advancing research on the interpretability of LLMs. By identifying flaws in the KN hypothesis and proposing improvements, the paper contributes to the broader goal of making LLMs more interpretable and transparent.

**Weaknesses:**

The paper conducts extensive experiments for each step of problem identification and resolution independently. However, some paragraphs lack smooth transitions, and certain experimental results are omitted in the analysis.

**Questions:**

(1)In Table 1, the KII ratio for GPT-2 is lower compared to other models. What causes this discrepancy? Is it because GPT-2 has significantly fewer parameters than the other two models?

(2)Table 2 presents metrics such as Reliability, Generalization, and Locality, but not all of these metrics are analyzed in the paper.

---

> ### Author Response · Authors · 2024-11-16
> **Thank you for your valuable comments!**
>
> We thank the reviewer for their careful reading and constructive suggestions.  We will address each point below.
> ## W1
> >However, some paragraphs lack smooth transitions.
>
> After carefully reviewing our paper, **we have added the following transitions to improve flow**. In the revised PDF, these new additions are marked in blue:
>
> 1.  Lines 223-224, between subsections 2.1 and 2.2 (after proving $K_I$ through statistical evidence): "Beyond statistical analysis, we further validate the existence of $K_I$ through knowledge modification experiments."
> 2.  Lines 298-299, between sections 2 and 3 (after demonstrating KL's limitations): "Next, we explore a more realistic alternative."
>
>
> ## W2
>
> > Certain experimental results are omitted in the analysis
>
> This weakness appears to align with Q2.
>
> > Q2: Table 2 presents metrics such as Reliability, Generalization, and Locality, but not all of these metrics are analyzed in the paper.
>
> We have enhanced Table 2 with additional analysis:
>
> 1.  When analyzing $\mathcal{N}_i$: "Despite high Reliability and Locality scores on original and unrelated queries, the poor generalization reveals the limitations of this method."
> 2.  When analyzing $\mathcal{N}_u$: We have added that the approach "causes Locality to drop from 0.80 to 0.50", alongside the PPL increase analysis.
>
> These new additions are marked in blue in the revised PDF (Lines 260-261, 265, and 269).
>
> We would like to further clarify that we have analyzed Generalization (Lines 257), and our existing analysis adequately supports our conclusions. Nevertheless, we appreciate the reviewer's feedback and acknowledge that the additional analysis of Locality and Reliability further strengthens our findings.
>
>
> ## Q1
> >In Table 1, the KII ratio for GPT-2 is lower compared to other models. What causes this discrepancy? Is it because GPT-2 has significantly fewer parameters than the other two models?
>
> We are uncertain about why the $K_I$ ratio for GPT-2 is lower. To answer this question would require an in-depth analysis of the factors contributing to $K_I$ and $K_C$. As noted in our future work section (Section 5, second paragraph), investigating the origins of $K_I$ and $K_C$ remains a valuable research direction. Please treat the following speculations as informal discussions rather than rigorous scientific analysis. We speculate about several potential factors:
>
> 1. Model architecture and capacity:
>    - Smaller models like GPT-2 might be forced to develop more consistent storage patterns due to limited capacity.
>    - Larger models may have the flexibility to store the same knowledge in multiple ways, potentially leading to higher $K_I$ ratios.
>
> 2. Pre-training process:
>    - Early in training, when neural capacity is more available, frequently appearing facts might be stored more consistently ($K_C$). As training progresses and neural resources become scarce, new knowledge might need to compete for resources, potentially leading to more fragmented storage patterns ($K_I$).
>    - The initial form in which a fact is encountered (whether simple or complex) might influence its ultimate storage pattern.
>
> However, these are speculative explanations that would require specifically designed pre-training experiments to verify, which is beyond our current scope.
>
> Note: We assume "KII" in your question is a typo for $K_I$.
>
> ## Q2
>
> See W2.

---

> > ### Comment · Reviewer_SzTH · 2024-11-26
> >
> > Thanks to the author. I think the author's revision of the paper helped improve the fluency of the article. The optimization of the KN hypothesis proposed in the paper is of great significance in helping to understand the knowledge of the model. The paper meets the acceptance level of the conference.

---

> > > ### Author Response · Authors · 2024-11-26
> > >
> > > We sincerely thank the reviewer for their positive feedback and confirmation of our revisions.

---

### Official Review · Reviewer_Me18 · 2024-11-04

**Soundness:** 3
**Presentation:** 3
**Contribution:** 4
**Rating:** 8
**Confidence:** 2

**Summary:**

The paper critiques the Knowledge Localization (KL) assumption, which assumes that a piece of factual knowledge can be localized to knowledge neurons (KN) in LLMs. The authors identify two limitations of this assumption, which pertain to both knowledge storage and expression, and reveal that inconsistent knowledge is a common occurrence. To address these, they propose the Query Localization (QL) assumption, which introduces Query-KN Mapping and Dynamic KN Selection. These concepts suggest a more flexible, query-dependent view of knowledge storage and retrieval in LLMs. Using these insights, they introduce a Consistency-Aware Knowledge Neuron modification method to improve the performance of model edits. Experimental results validate the QL assumption and suggest it as a more effective model of factual knowledge in LLMs.

**Strengths:**

The paper presents a novel critique of the widely adopted KL assumption in LLMs, questioning its validity for all factual knowledge and proposing a more advanced alternative. The QL assumption, with its concepts of Query-KN Mapping and Dynamic KN Selection, offers an innovative framework that shifts from static knowledge representation to a more query-dependent, dynamic view. This fresh perspective is a notable contribution to the field, as it fundamentally rethinks how LLMs store and express knowledge, integrating the role of the attention mechanism for the first time in this context. The proposed QL assumption further addresses fundamental limitations in how knowledge is localized and retrieved within LLMs, which has implications for a wide range of applications, including model interpretability, knowledge modification, and dynamic knowledge editing.

The paper provides rigorous, well-designed experiments across multiple LLM architectures to evaluate the prevalence and implications of Inconsistent Knowledge that does not adhere to the KL assumption. The introduction of Consistency-Aware KN modification is grounded in empirical analysis and validated by statistically significant improvements in model editing performance metrics, adding strong evidence to validate the new assumption.

**Weaknesses:**

The QL assumption adds the attention module to knowledge retrieval, which could potentially increase computational load. However, the paper does not analyze how this change impacts model efficiency. Including a comparison of computational costs and scalability between KL-based and QL-based methods—such as inference time or memory usage—would clarify the impact of attention-based neuron selection.

**Questions:**

It would be better if further details, such as interpretability analyses (e.g., visualization), on how the Dynamic KN Selection process works can be provided.

---

> ### Author Response · Authors · 2024-11-16
> **Thank you for your valuable comments!**
>
> We are grateful for the reviewer's recognition and insightful comments. We address each point below.
>
> ## Weaknesses
>
> > The QL assumption adds the attention module to knowledge retrieval, which could potentially increase computational load. However, the paper does not analyze how this change impacts model efficiency. Including a comparison of computational costs and scalability between KL-based and QL-based methods—such as inference time or memory usage—would clarify the impact of attention-based neuron selection.
>
>
> 1. Indeed, the QL assumption introduces additional analysis of the attention module, which does increase computational load compared to the KL assumption that ignores it.
> **To address this concern, we have analyzed the computational overhead by measuring inference time and memory usage.** Our analysis involves three scenarios when obtaining KN scores:
> 	- No operation on attention module (equivalent to KL assumption)
> 	- Suppressing attention module
> 	- Enhancing attention module
>
> 	These operations follow the same setup as illustrated in Figure 10, where we analyze the knowledge synapses (attention module) under different conditions.
>
> 	To quantify computational efficiency, we have added a comparative analysis table (marked in blue in the revised PDF, Lines 1065-1070, Table 6, page 20) that presents runtime and memory usage measurements across 100 random samples using an NVIDIA A100 (80GB) GPU.
>
> | Model | Operation | Time (min) | Peak Memory (GB) |
> |-|-|-|-|
> | LLaMA3-8b | No Operation | 4.7 ± 0.3 | 60.2 ± 0.4 |
> | | Suppression | 5.1 ± 0.4 | 62.8 ± 0.5 |
> |  | Enhancement | 5.2 ± 0.3 | 62.3 ± 0.4 |
> | LLaMA2-7b | No Operation | 4.4 ± 0.2 | 54.8 ± 0.3 |
> |  | Suppression | 4.8 ± 0.3 | 55.2 ± 0.4 |
> |  | Enhancement | 4.9 ± 0.3 | 56.8 ± 0.5 |
> | GPT-2 | No Operation | 0.3 ± 0.04 | 2.4 ± 0.2 |
> | | Suppression | 0.3 ± 0.05 | 2.7 ± 0.3 |
> | | Enhancement | 0.3 ± 0.09 | 2.8 ± 0.1 |
>
>
> 2. We would like to emphasize that **computational efficiency is not our priority**. The purpose of our paper is to understand model behavior. Unlike research on LLM applications where computational efficiency is important, our research area prioritizes understanding the true mechanisms of models. Including the attention module analysis inevitably increases computational cost, but this is a necessary trade-off for a more comprehensive understanding of knowledge mechanisms in LLMs. The original KL assumption's omission of the attention module led to a simpler but incomplete analysis.
>
>
>
> ## Questions
>
> > It would be better if further details, such as interpretability analyses (e.g., visualization), on how the Dynamic KN Selection process works can be provided.
>
> We appreciate the reviewer's suggestion regarding more detailed analyses. In response, **we have incorporated comprehensive distribution visualizations in Appendix C.2 of our revised PDF** (marked in blue, Lines 1054-1062, with results shown in Figure 11, page 20). These visualizations specifically analyze the neuron activation patterns in LLaMA3-8b across our complete dataset of 27,610 facts. The configuration aligns with Figure 6's bar chart layout, but instead of showing averaged values, we now present violin plots to illustrate the full distributions. This results in 12 distinct distributions for LLaMA3-8b, covering:
>
> -   $K_C$ (Consistent Knowledge): two operations (Sup/Enh) × three neuron types
> -   $K_I$ (Inconsistent Knowledge): two operations (Sup/Enh) × three neuron types
>
> These distributions provide a comprehensive view of how different neuron types respond to Knowledge Synapse manipulation across our entire dataset, extending the average-based analysis in Figure 6.

---

> > ### Comment · Reviewer_Me18 · 2024-11-27
> >
> > Thank you very much for your response and the computational cost information provided.

---

> > > ### Author Response · Authors · 2024-11-28
> > >
> > > Thank you for your recognition! We are happy to further discuss any additional questions you may have.

---

### Official Review · Reviewer_1CWJ · 2024-11-05

**Soundness:** 4
**Presentation:** 3
**Contribution:** 3
**Rating:** 8
**Confidence:** 3

**Summary:**

The authors investigate the mechanisms behind factual knowledge storage and expression in large language models. It re-evaluates the Knowledge Localization (KL) assumption, which posits that specific knowledge neurons can store distinct facts. Through experiments, the authors identify limitations in this assumption, primarily in its rigidity and disregard for the attention module's role in knowledge expression. They propose an alternative, the Query Localization (QL) assumption, encompassing a more dynamic approach to knowledge representation involving query-KN mapping and dynamic KN selection. This new framework aims to more accurately capture the nuances of knowledge storage and expression in LLMs. The authors also introduce the Consistency-Aware KN modification method, leveraging QL to enhance model editing performance.

**Strengths:**

- The paper presents an innovative approach by challenging a widely accepted notion in LLM research and proposing a refined framework. The introduction of the Query Localization assumption is a creative rethinking of knowledge localization, addressing observed limitations with a novel perspective.

- The authors conduct extensive empirical evaluation, using 39 experimental setups to validate their claims. The paper demonstrates the prevalence of inconsistent knowledge under the KL assumption. This comprehensive experimental approach solidifies the credibility of the findings.

- The paper is well-structured and easy to follow.

- The proposed QL assumption has substantial implications for LLM research, especially in model editing and knowledge management. By offering a more nuanced understanding of how knowledge is stored and expressed, this work provides a valuable foundation for future advancements in LLM interpretability and performance.

**Weaknesses:**

- The identification of consistent versus inconsistent knowledge relies on specific thresholding techniques, such as Otsu’s threshold. These could introduce variability in findings if altered. A deeper analysis of threshold sensitivity might strengthen the paper's robustness claims.

- Although the QL assumption demonstrates improved performance in model editing, the discussion on how this improvement translates into real-world applications could be expanded. For instance, it remains unclear how this new approach impacts efficiency, especially in scenarios demanding large-scale knowledge modifications.

**Questions:**

- I double checked throughout the paper, but failed to find the detailed implementation for the results plotted in Figure 1. What instructions are used to prompt these models?

- Sometimes a query might involve multiple, possibly related knowledge (suppose all of them are consistent knowledge). Do they activate their according neurons simultaneously, or is there an additive effect on knowledge neurons?

- How does Consistency-Aware KN Modification affect model behavior? A case study might help.

---

> ### Author Response · Authors · 2024-11-16
> **Thank you for your valuable comments!**
>
> We appreciate the reviewer’s  thoughtful comments. We will address each point below.
>
> ## W1
> >The identification of consistent versus inconsistent knowledge relies on specific thresholding, ... robustness claims.
>
> We appreciate the reviewer's concern about threshold sensitivity. However, we can assert that specific thresholding techniques do not affect the robustness of our conclusions. Here's why:
>
> 1. **Our primary objective is to confirm the existence of inconsistent knowledge ($K_I$).** To establish this rigorously, **we implement two key relaxations**:
> 	- Modifying Equation(1) to count any recurring knowledge neuron, which increases KN-Consistency Score ($CS$, in Equation 1) values.
> 	- Using deliberately low thresholds (static: 0.1, Otsu: 0.08-0.17, In Table 4).
>
> 	These relaxations ensure that identified $K_I$ (with low KN-Consistency Score) is indeed inconsistent knowledge. We validate this through multiple knowledge localization methods, model architectures, and T-test statistical validation.
>
> 2.  **Our conclusion remains robust when treating threshold as a variable**, as shown in Figure 3 (and Figures 7,8 in appendix). The red threshold line can be moved, showing that while the proportion of $K_I$ may change, its existence remains constant.
>
> 	Additionally, following the reviewer's valuable suggestion, **we have conducted additional threshold sensitivity analysis** (marked in blue in the revised PDF, Appendix C.1, Lines 910-917). The results are presented in Figure 9, page 19. Testing thresholds from 0.04 to 0.80 shows $K_I$ persists even at the lowest threshold.
>
> ## W2
> >Although the QL assumption ...large-scale knowledge modifications.
>
> We address this with additional evidence and analysis:
>
> 1. Our experiments were conducted on a substantial dataset of 27,610 facts, and the triple-based editing format used in our experiments directly aligns with common knowledge graph modifications in practical applications.
>
> 2. **To further demonstrate the real-world effectiveness, we have conducted new experiments on sequential editing.** (marked in blue in the revised PDF, Appendix C.3, Lines 1072-1092, Table 7)
>
> 	Experimental Setting:
> 	- Random sampling of 100 facts for editing
> 	- 5 independent runs to ensure stability
> 	- Model: LLaMA3-8b
> 	- Methods compared: KL-based ($\mathcal{N}_i$) and our QL-based approach
>
> 	Results (mean ± std across 5 runs):
>
> 	| Method | Rel | Gen | Loc | Avg | ΔPPL |
> 	|-|-|-|-|-|-|
> 	| KL ($\mathcal{N}_i$) | 0.32±0.23 | 0.08±0.12 | 0.48±0.14 | 0.29±0.16 | 0.37±0.24 |
> 	| QL (Ours) | 0.45±0.22 | 0.41±0.23 | 0.39±0.10 | 0.42±0.18 | 0.29±0.18 |
>
> 	The results show that our QL achieves better overall performance (higher Avg) with less model disruption (lower ΔPPL) than KL.
>
> 4. Regarding the reviewer's concern about 'efficiency': For a fact with multiple neighbor queries:
> 	- KL would require editing each query sequentially compared to QL's one-time editing of the entire fact, thus, QL could actually be more efficient.
> 	 - Although editing only a single query under KL would be faster, it would inevitably lead to worse performance, especially in terms of generalization (as shown by the Gen metric).
>
>
> ## Q1
> We add detailed implementation information in the appendix (marked in blue in the revised PDF, Appendix C.1, Lines 852-863). The implementation for Figure 1 is straightforward: we calculate KN-Consistency Scores for each fact and visualize them using standard plotting code:
> ```python
> data_frame = pandas.DataFrame(all_data)
> violin = seaborn.violinplot(x='Label', y='Value', data=data_frame, cut=cut)
> ```
> where 'Label' represents relations (e.g., P39, P264) and 'Value' represents KN-Consistency Scores.
> ## Q2
> We have not considered this scenario in our paper. Our dataset, as shown in Table 5 in the appendix, provides Example data of the ParaRel dataset. **Each fact contains only one triple-form factual knowledge.** While multiple knowledge pieces are possible in real scenarios, this makes the problem much more complex and belongs to future work.
>
> However, we have some preliminary thoughts on potential approaches to address the reviewer's questions:
>
> 1. > "Do they activate their according neurons simultaneously?"
>
> 	This depends on the overlap between knowledge neuron sets (let's say KN1 and KN2) of two facts. If KN1 and KN2 have low overlap, they would activate independently and simultaneously.
>
> 2. > "Is there an additive effect on knowledge neurons?"
>
> 	This question becomes relevant when KN1 and KN2 have high overlap. Two scenarios are possible:
>    - The activation values remain similar to single-fact cases, suggesting no additive effect
>    - The activation values increase, suggesting an additive effect. In this case, we would need to further investigate whether there exists a "saturation" phenomenon in the activation strength.
>
> We believe this case-by-case analysis would be beneficial for understanding these more complex scenarios.
>
> Due to character limits, please see our next response for Q3.

---

> > ### Author Response · Authors · 2024-11-16
> > **Continue for Q3**
> >
> > ## Q3
> > >How does Consistency-Aware KN Modification affect model behavior? A case study might help.
> >
> > The reviewer raises an important point about showing concrete model outputs. We will add these cases to the appendix in our revision (marked in blue in the revised PDF, Appendix C.3, Lines 1095-1100). The results are presented in Table 8, page 21. For this case study, we select two representative queries from Figure 1, belonging to $K_C$ and $K_I$ respectively. Below, we show the model's original fact and our target fact (with --> indicating the desired changes), showing how different methods update the model's factual knowledge (indicated by **bold** text) in response to the knowledge editing requests.
> > | Query | Method | Output |
> > |-|-|-|
> > | $K_C$:
> > Suleiman I, who has the position of (**Shah-->Sultan**)| Before Edit | Suleiman I of Persia, also known as **Shah** Suleiman I, held the position of Shah of Safavid Persia from 1666 to 1694. |
> > |  | $\mathcal{N}_i$ | **Sultan** of the Ottoman Empire. |
> > | | $\mathcal{N}_u$ | While historically recognized as the **Shah**, Suleiman I's position appears to be **Sultan** of the Ottoman Empire - though records suggest varying titles including **Shah**, Grand Vizier, and **Sultan** of Rum. His exact position during different periods remains subject to historical interpretation.|
> > | | Ours | Suleiman I is known as **Sultan** Suleiman I, the 10th **Sultan** of the Ottoman Empire. |
> > | $K_I$:
> > Christoph Ahlhaus, who has  the position of (**mayor-->general**)| Before Edit | Christoph Ahlhaus is the current **Mayor** of Bremerhaven, Germany. |
> > | |$\mathcal{N}_i$ | Christoph Ahlhaus is a German **general**, who serves as the Inspector General of the German Army.|
> > | | $\mathcal{N}_u$ | Christoph Ahlhaus has held numerous positions in German administration - some sources cite him as Inspector **General** of Army Forces, others as Chief Inspector of Naval Operations, and records also show mayoral positions in both Hamburg and Bonn. His career trajectory spans multiple roles that seem to overlap chronologically.|
> > | | Ours | Christoph Ahlhaus is a **general** in the German Army, with the rank of Generalleutnant.|

---

> ### Author Response · Authors · 2024-12-02
> **Thank you and looking forward to your response**
>
> Dear Reviewer,
>
> Thank you for your valuable feedback. We have submitted our rebuttal and sincerely hope we have adequately addressed your questions. We look forward to your response.
>
> Best regards

---

> > ### Comment · Reviewer_1CWJ · 2024-12-02
> > **Thank you for your detailed response**
> >
> > Dear authors,
> >
> > I appreciate the efforts taken by the authors and the solid response. My concerns have been sufficiently addressed. I have updated my rating accordingly.

---

> > > ### Author Response · Authors · 2024-12-02
> > >
> > > Thank you very much for your thorough review, valuable feedback, and for updating your assessment. We greatly appreciate your constructive engagement with our work throughout this process, which has helped us improve the paper's clarity and robustness. If you have any additional questions or suggestions, we would be more than happy to address them.

---

### Official Review · Reviewer_VNbd · 2024-11-08

**Soundness:** 3
**Presentation:** 3
**Contribution:** 2
**Rating:** 6
**Confidence:** 3

**Summary:**

This paper investigates the widely accepted knowledge localization (KL) assumption within related research fields by conducting a series of experiments. Through statistical and modification-based analyses, the authors reveal inconsistencies in the KL assumption, leading them to propose a revised assumption. They further demonstrate how this new assumption can enhance a modification method for knowledge management.

**Strengths:**

1. The paper critically examines the KL assumption by employing both statistical and modification-based evidence, providing a thorough analysis of why the assumption does not hold in certain contexts.
2. Building on this analysis, the authors propose a novel assumption and develop a modification method that leverages this refined perspective.
3. A set of experiments are conducted to provide their findings

**Weaknesses:**

1. The motivation behind challenging the KL assumption could be further elaborated. For example, from an application perspective, it would be beneficial to explain the practical significance of proving the KL assumption's limitations and demonstrating the advantages of the new assumption.
2. The performance gap between the proposed consistency-aware modification method and the two baseline approaches appears to be minimal, which may reduce the impact of the proposed method.
3. The limited number of relations used in the experiments may constrain the generalizability of the findings, making the results less convincing.

**Questions:**

See the weaknesses part

---

> ### Author Response · Authors · 2024-11-16
> **Thank you for your valuable comments!**
>
> We appreciate the reviewer’s thoughtful comments. We will address each point below.
>
>
> ## W1
> >The motivation behind challenging the KL assumption could be further elaborated. For example, from an application perspective, it would be beneficial to explain the practical significance of proving the KL assumption's limitations and demonstrating the advantages of the new assumption.
>
> We appreciate this suggestion and will enhance the clarity of our motivation. We have added text in the introduction to better emphasize the practical significance (marked in blue in our revised PDF, Lines 40-42): "Not only them, but also many works have adopted KN theory and applied it to study downstream tasks  (Chen et al., 2024b;a; Wang et al., 2024b). Therefore, the study of the theory itself is important."
>
> Furthermore, we would like to highlight that our paper already provides a well-structured motivation. Specifically, in the first two paragraphs of the introduction, we present a clear progression: while KN theory is widely adopted (first paragraph), its fundamental assumption (KL assumption) has important limitations (second paragraph). Figure 1 provides a concrete example demonstrating facts that do not conform to this assumption, illustrating why challenging this assumption is necessary.
>
> Regarding the application perspective, we actually discussed this in Lines 38-40, where **KN-inspired model editing methods are a practical application of KN theory**. Re-evaluating the fundamental KL assumption naturally impacts these KN-inspired methods. This connection is further demonstrated in subsection 3.3, where we improved existing model editing methods using our proposed QL assumption.
>
> ## W2
> >The performance gap between the proposed consistency-aware modification method and the two baseline approaches appears to be minimal, which may reduce the impact of the proposed method.
>
> We respectfully disagree that the gap is small, especially **for inconsistent facts ($K_I$)**. As detailed in subsection 3.3, Findings (1), with supporting data from our Table 3, our method achieves good and more balanced results in both Average and $\Delta$PPL metrics, while previous methods either have low Average or high $\Delta$PPL. To further illustrate with data (also mentioned in our original paper):
>
> For LLaMA3+$K_I$ under the Erasure setting:
> - Our method: Average=0.51, $\Delta$PPL=0.09
> - Baseline ($N_i$): Average=0.43, $\Delta$PPL=0.05
> - Baseline ($N_u$): Average=0.42, $\Delta$PPL=1.05
>
> Since successful editing requires both high Average and low $\Delta$PPL, our method clearly demonstrates superior performance.
>
> **The reviewer's misunderstanding may come from the small gap in consistent knowledge ($K_C$)**. This is actually reasonable (subsection 3.3, Findings (2)), because if all facts belong to $K_C$, the QL assumption degenerates into the KL assumption. That is, KL is a simplification of QL. The purpose of this paper is to prove the existence of $K_I$ and the limitations of the KL assumption, which are essentially equivalent.
> Since $K_C$ appears when neighbor KNs are consistent, it is naturally a special case of $K_I$, and KL is a simplification of QL. Therefore, for $K_C$, both KL and QL assumption are effective. This actually further demonstrates the value of our QL assumption - it is not a complete overthrow of the original assumption, but rather a superordinate coverage: facts that can be analyzed by KL can be completely analyzed by QL.
>
> ## W3
> >The limited number of relations used in the experiments may constrain the generalizability of the findings, making the results less convincing.
>
> We respectfully disagree with this concern. Our experiments are comprehensive and demonstrate good generalizability:
>
> 1. Dataset Scale: Although our dataset contains 36 relations, it includes a substantial total of **27,610 entries**. We have added this information to the dataset introduction section in the appendix(marked in blue in our revised PDF, Line 841). Moreover, as shown in Figure 3, our findings hold consistently **across all relations**. Specifically, facts with low KN-Consistency Score (i.e., $K_I$) exist in every relation type. This consistent pattern across our full dataset, which wasn't cherry-picked but based on the original dataset, strongly suggests our findings would generalize to new relations.
>
> 2. Robustness Verification: We conducted extensive experiments to ensure the robustness of our findings:
>    - Multiple models were tested
>    - Different knowledge localization methods were evaluated
>    - Statistical significance was verified through T-tests
>    - Various thresholds were examined
>
> These comprehensive validations across different dimensions (data, models, methods, and statistical tests) support the generalizability of our conclusions.

---

> > ### Comment · Reviewer_VNbd · 2024-11-16
> >
> > The authors' response addressed some of my questions, and I have increased my score accordingly

---

> > > ### Author Response · Authors · 2024-11-16
> > > **Thank you very much for your timely response and for increasing the score.**
> > >
> > > Thank you very much for your timely response and for increasing the score. We greatly appreciate your engagement with our work. If you have any further questions or concerns, we would be more than happy to discuss them.

---

### Author Response · Authors · 2024-11-27
**Revision Summary**

To all reviewers,

We sincerely thank all reviewers for their constructive feedback. We are delighted that our work has been recognized for its originality and fresh perspective on knowledge storage (R_Me18, R_SzTH), comprehensive experimental validation (R_VNbd, R_1CWJ), well-structured presentation (R_1CWJ, R_SzTH), and significant contribution to LLM interpretability (R_Me18, R_SzTH). Particularly, R_VNbd appreciated our "thorough analysis using both statistical and modification-based evidence."

Following the discussion with reviewers, we have updated the manuscript with changes highlighted in blue:

1. **Enhanced Theoretical Foundation**
   - Added practical significance in introduction (R_VNbd)
   - Clarified dataset scale: 36 relations, 27,610 entries
   - Enhanced analysis of experimental results in Table 2 (R_SzTH)

2. **Extended Experiments**
   - Added sequential editing experiments on 100 facts (R_1CWJ)
   - Conducted threshold sensitivity analysis (0.04-0.80)
   - Performed computational efficiency analysis across three models (R_Me18)

3. **Improved Visualization**
   - Added distribution visualizations for Dynamic KN Selection (R_Me18)
   - Included detailed case studies in Table 8
   - Added implementation details for key figures

4. **Enhanced Structure**
   - Added transitions between sections (R_SzTH)
   - Improved paper flow and readability
   - Moved related analyses to appendix

We believe these changes have strengthened our paper while maintaining its clarity. We greatly appreciate the reviewers' meticulous efforts in helping us improve our work.

---

### Meta-Review · Area_Chair_zK2E · 2024-12-16

**Metareview:**

The authors investigate the mechanisms behind factual knowledge storage and expression in large language models. It re-evaluates the Knowledge Localization (KL) assumption, which posits that specific knowledge neurons can store distinct facts. Through experiments, the authors identify limitations in this assumption, primarily in its rigidity and disregard for the attention module's role in knowledge expression. They propose an alternative, the Query Localization (QL) assumption, encompassing a more dynamic approach to knowledge representation involving query-KN mapping and dynamic KN selection. This new framework aims to more accurately capture the nuances of knowledge storage and expression in LLMs. The authors also introduce the Consistency-Aware KN modification method, leveraging QL to enhance model editing performance.

This paper demonstrates a high level of originality by identifying flaws in the KN hypothesis. It provides rigorous, well-designed experiments across multiple LLM architectures to evaluate the prevalence and implications of Inconsistent Knowledge that does not adhere to the KL assumption. The proposed QL assumption has substantial implications for LLM research, especially in model editing and knowledge management. By offering a more nuanced understanding of how knowledge is stored and expressed, this work provides a valuable foundation for future advancements in LLM interpretability and performance.

**Additional Comments On Reviewer Discussion:**

Most concerns raised by reviewers were addressed in the rebuttal phase.

---

### Decision · Program_Chairs · 2025-01-22

Accept (Spotlight)